# Evaluating the Impact of Task Aggregation in Workflows with Shared Resource Environments: use case for the MONARCH application

Manuel G. Marciani[1,2], Miguel Castrillo[1], Gladys Utrera[2,1], Mario C. Acosta[1], Bruno P. Kinoshita[1], and Francisco Doblas-Reyes[1,3]

[1]Barcelona Supercomputing Center (BSC), Plaça Eusebi Güell, 1-3, Barcelona,Spain
[2]Universitat Politècnica de Catalunya (UPC), Carrer Jordi Girona, 1-3, Barcelona, Spain
[3]Institució Catalana de Recerca i Estudis Avançats (ICREA), Barcelona, Spain

**Correspondence:** Manuel G. Marciani (manuel.gimenez@bsc.es) and Mario C. Acosta (mario.acosta@bsc.es)

**Abstract.** High Performance Computing (HPC) is commonly employed to run high-impact Earth System Model (ESM) simulations, such as those for climate change. However, running workflows of ESM simulations on cutting-edge platforms can take a long time due to the congestion of the system and the lack of coordination between current HPC schedulers and workflow manager systems (WfMS). The Earth Sciences community has estimated the time in queue to be between 10% to 20% of the runtime in climate prediction experiments, the most time-consuming exercise. To address this issue, the developers of Autosubmit, a WfMS tailored for climate and air quality sciences, have developed wrappers to submit multiple subsequent workflow tasks – the indivisible unit of compute as defined by the workflow – in a single remote job, without altering any of the tasks. However, although wrappers are widely used in production for community models such as EC-Earth3, MONARCH, and Destination Earth simulations, to our knowledge, the benefits and potential drawbacks have never been rigorously evaluated. Later, the developers of Autosubmit noticed that the impact of using wrappers was related to the past utilization of the user and its group, which the popular scheduler Slurm uses to compute the priority of the queueing jobs. In Slurm's methodology, this past utilization is quantified in the fair share factor. The objective of this paper is to quantify the impact of wrapping subsequent tasks on queue time and understand its relationship with the fair share and the job's CPU and runtime request. To do this, we used a Slurm simulator to reproduce the behavior of the scheduler and, to recreate a representative usage of an HPC platform, we generated synthetic static workloads from data of the LUMI supercomputer and a dynamic workload from a past flagship HPC platform. As an example, we introduced jobs modeled after the MONARCH air quality application in these workloads, and we tracked their queue time. We found that, by simply joining tasks, the total time-to-solution of the simulation reduces up to 7% with respect to the runtime of the simulations, and we believe that this value is larger the longer the workflow, since longer wrappers could be created and hence less jobs are submitted to the scheduler. This saving translates to absolute terms of about eight days less wasted in queue time for half of the simulations from the IS-ENES3 consortium of CMIP6 simulations. We also identified in the static experiments a high inverse correlation of $-0.87$, between the queue time and the fair share factor.

# 1  Introduction

High-impact Earth System Model (ESM) simulations are normally executed in cutting-edge High Performance Computing (HPC) resources. In these environments, they typically entail several steps, including model compilation, data movement, model execution, and post-processing. Furthermore, the execution of the model is often divided into multiple segments, which we refer to as "chunks" of simulated time, which is done to provide natural checkpoints for the simulations and to comply with maximum execution time constraints imposed by the system administrators (i.e., the wallclock). This results in workflows of up to thousands of individual tasks – the workflow's indivisible unit of compute –, as is the case with Destination Earth digital twin's (Hoffmann et al., 2023). To handle all of this complexity, ESM simulations are automated with workflows.

Given the high cost associated with running such simulations, the Earth Sciences community has always been looking for ways of reducing the time-to-solution, which is the total time spanning from the first submission to the finish of the latest. The traditional front is to minimize the execution time of the most computationally intensive code within the model, such as the work of Irrmann et al. (2022).

However, lately the community has drawn attention to the efficiency of the simulations, taking into account not only the runtime of the simulation but also the time spent with the postprocessing, failures, and in the queue. With this in mind, Balaji et al. (2017) proposed a set of performance metrics for Earth system model simulations. Among these metrics, the authors proposed the simulated years per day (SYPD), which is the ratio of the time simulated in years with respect to the runtime of the simulation in days, as well as the actual simulated years per day (ASYPD), which is the simulated time in years divided by the the time-to-solution of the simulation, therefore also accounting for time in queue and failures.

In Acosta et al. (2024), the authors computed these metrics for 33 CMIP6 simulations executed on 14 machines. Their analysis showed that the difference between ASYPD and SYPD ranged from 0% to 78%. But, they noted that not all institutions reported ASYPD consistently. Some accounted for both interruptions and queue time, while others accounted only for queue time. For those institutions that only accounted for queue time, the spread was between 10% and 20%. The authors therefore concluded "that queuing time represents an increment of around 10%–20% of the speed of the ESM."

The software that handles the queue of resource requests in HPC platforms is called workload manager, or scheduler. We call each resource request a job. The development of schedulers is an active area of research within the HPC scheduler community, focusing on optimizing job scheduling policies to minimize queue time, while keeping a high utilization of the machine (Brucker, 2007). Some examples of these policies are: first-come, first-served, least processing time, and priority-based. This last one is called *multifactor* in the ubiquitous Slurm workload manager (Jette and Wickberg, 2023).

In Slurm, under the multifactor policy, jobs are ordered decreasingly by an integer called "priority", which is computed from "factors" that come from the job request ("size", "partition", "QoS"), time in queue ("age"), and the user's "fair share". The purpose of this last factor, which is typically prioritized by system administrators, is to balance the responsiveness of the machine by favoring users who have not used their quota of the resources while penalizing those who overuse the machine. Besides the job's priority, Slurm also aims to maximize system utilization by implementing a technique called backfill (Srini-

vasan et al., 2002). This technique allows jobs with less priority to use free spots in the machine, as long as they do not interfere with the jobs ahead of them.

Observing the impact of fair share on the priority, the developers of Autosubmit (Manubens-Gil et al., 2016), a Workflow Manager System (WfMS) designed for climate and air-quality applications, have implemented a task aggregation technique called "wrappers." The idea is that Autosubmit aggregates, or wraps, multiple subsequent tasks into a single job submission to save the users from queuing multiple jobs under a low priority. No changes are done to the tasks, therefore the natural checkpointing is still maintained, and the dependencies are ensured among the tasks. Additionally, wrapping is also employed for concurrent tasks in order to comply with the maximum number of jobs in the queue.

Aggregation was implemented in other fields, with different degrees of sophistication. In Earth Sciences, Mickelson et al. (2020) suggested using Cylc's feature to submit multiple jobs to reduce the queue time. Both Aiida (Huber et al., 2020) and Snakemake (Mölder et al., 2021) — from the material and life sciences fields, respectively — provide a way of submitting multiple workflow tasks as a single job. The former implements this via a "metascheduler" plugin (HyperQueue plugin), while the latter refers to aggregation as "grouping."

Moreover, pilot-job systems were developed to increase workflow throughput with more sophisticated solutions (Turilli et al., 2018). These systems are characterized by implementing "resource placeholders, multi-level scheduling, and coordination patterns to enable task-level distribution and parallelism on multi-tenant resources." One major example of a modern pilot-job system is Radical-PILOT (Merzky et al., 2021).

Wrappers share some of the features of a typical pilot-job system. Wrappers provide a simpler "resource placeholder," where all the task requests are added into a large submission. However, their objective is to either increase throughput by reducing submissions or to comply with maximum job restrictions. Therefore, besides fault tolerance, wrappers do not improve task scheduling and coordination within the allocation.

Currently, in our field, wrappers are used to reduce the queue time in the MONARCH (Klose et al., 2021) application, in the climate model EC-Earth3 (Döscher et al., 2022). They are also employed in the Destination Earth digital twin, to both reduce queue time and to comply with the platforms' policy regarding maximum jobs queuing per user.

Although aggregation is utilized by Autosubmit users and also in other fields, with positive impact reported, there is a lack of understanding of the reasons and conditions under which aggregating reduces queue time. Therefore, in this work, we tested whether wrapping subsequent tasks together reduces queue time and if the fair share is the most important factor in reducing queue time.

To test both theses, we measured queue time by simulating the HPC environment using a Slurm simulator (Ana Jokanovic, Marco D'Amico, and Julita Corbalan, 2018). We modeled a congested workload of an HPC platform and use the workflow of the MONARCH air-quality application. Then, we ran under the same conditions a simulation with wrappers and without.

We found that combining tasks into one submission reduces the queue time by at most 7% of the total workflow runtime, and we observe gains across all the tested fair share values. Moreover, this reduction can be greater for longer workflows due to the many times less tasks being submitted. In the context of the CMIP6 runs of the IS-ENES3, this reduction equates to at

least eight fewer days of queue time for half of the runs. Additionally, we found a high inverse correlation of -0.87 between the fair share factor and time in the queue.

Our study provides a quantification of the reduction of queue time achieved by aggregating tasks of air quality applications that use HPC resources on congested platforms. This approach can also be applied to other simulations with temporal constraints that require a large amount of resources for a sustained time. Our results help to advance the understanding, from the user side, of how to optimize the submission in order to reduce the total queue time of their workflows.

This paper is organized as follows: the background section 2 defines wrapping and introduces major points of the scheduling policy of Slurm. The methodology 3 provides a detailed explanation of the experimentation. This includes how we introduce the workflow onto the workload files, how we control the fair share, and the software we used and implemented to run the simulations. The results section 4 exposes the output of the experiments. This leads to the discussion 5 chapter, where we comment our findings and the relation between fair share and queue time. Finally, in the conclusions, we evaluate if our findings support our theses.

## 2 Background

In this section, we start with an overview on Slurm's scheduling in subsection 2.1 because of its wide adoption across HPC platforms. Then, we explain why and how wrappers are used in subsection 2.3.

### 2.1 Scheduling Algorithms of Slurm

When a job is submitted, Slurm immediately attempts to allocate resources to it. If no resources are available, the job will be placed in the queue. This queue is sorted in descending order according to the priority integer, which updates periodically.

Slurm's scheduling design consists of two coordinated algorithms that traverse the ordered list of queued jobs and attempt to allocate resources to them. The main algorithm tries to schedule as many jobs as possible from the queue. If it fails because of a lack of resources, it breaks the loop and sleeps. There are two options for the second algorithm: built-in and backfill. The first option works like the main algorithm. The second option is the backfill algorithm (Srinivasan et al., 2002).

The backfill algorithm is an optimization technique that allows jobs with lower priority to be scheduled before higher-priority jobs, as long as they do not interfere with the start time of the higher-priority jobs. Therefore, the smaller the job is in terms of wallclock and/or CPUs requested, the more likely it is to be backfilled. Without considering the effects of the backfill algorithm, jobs with higher priority are scheduled earlier.

Slurm uses the multifactor policy by default to compute the priority integer. In Slurm terminology, factors are always floats stored in double-precision between zero and one. The priority of the job is computed as a weighted sum of the following factors: age, size, association, fair share, and quality of service (QoS). A 32-bit encoded integer weight is associated with each of these values.

The age factor, which is the one related to the time in queue, starts at zero and linearly grows until reaching the maximum, which is configured by setting the flag *PriorityMaxAge* in the Slurm configuration file. The default is seven days. In Equation

1 we have the expression used to compute the factor where $t$ is the time elapsed from submission, discounted by the time voluntarily held by user and waiting for its dependencies to finish, and $T$ is the total amount of seconds set by the administrator with flag *PriorityMaxAge*.

$$a_i(t) = \begin{cases} \frac{t}{T}, t \leq T \\ 1, t \geq T \end{cases}. \tag{1}$$

By default, the size factor is the proportion of CPUs used by the job with respect to the total available. Therefore, when the whole machine is requested, the maximum is reached. System administrators can invert this by setting the flag *PriorityFavorSmall* to true, which would give job requesting a single CPU the maximum value.

The size factor is computed as in Equation 2, where $r_i$ is the number of CPUs requested by job $i$ and $S$ is the total number of CPUs on the machine.

$$s_i = \frac{r_i}{S} \quad \text{or} \quad s_i = \frac{S - r_i + 1}{S}. \tag{2}$$

QoS stands for quality of service. It normally used by system administrators to benefit smaller jobs (such as debugging jobs), since by default Slurm gives more priority to larger jobs. From the user side, this means higher priority upon submission, hence less waiting time, at the cost of a more constrained job submission. System administrators typically create different QoSs according to job length and size – for example, standard, debug, xlarge, and xlong.

Each QoS is configured with an associated priority, which is another integer value. Then the factor is computed as the proportion with respect to the largest priority of all QoSs. Therefore, if the job $i$ is submitted to QoS $u$ with priority $q_u$, its QoS factor is calculated according to Equation 3, where $Q$ is the set of the priorities of all QoSs configured by the system administrator.

$$q_i = \frac{q_u}{max_{q \in Q} q}. \tag{3}$$

The fair share factor is the one that aims to balance the resources among users. It is computed in two different ways, which will be addressed in its separate Section 2.2.

Finally, the job's priority is computed according to Equation 4. $P_i(t)$ the priority of job $i$'s at time $t$. $a_i$ is the age factor and $w_a$ is its weight, $s_i$ is the size factor and $w_s$ its weight, $f_i$ is the fair share of the user that submitted job $i$ and $w_f$ is the fair share weight, and finally $q_i$ and $w_q$ are the QoS factor and its weight.

$$P_i(t) = a_i(t) \cdot w_a + s_i \cdot w_s + f_i \cdot w_f + q_i w_q. \tag{4}$$

## 2.2 Fair Share Factor

The fair share factor is the quantification of a user's right to the machine, which is used to balance the responsiveness of the machine among all users according to their entitlement to it.

There are two ways to calculate the fair share factor: the *Classic* method and the *Fair Tree* method (SchedMD, 2019). We will focus on the Fair Tree method because it is the one used by default.

In the Fair Tree algorithm, users are associated with an account, which is typically created for each project. Accounts can also be associated to an account. Both users and accounts are given an integer value called *RawShare*, usually related to their budget for the machine. Then, the usage of both user and account is tracked via the *RawUsage* integer. Normally, the *RawUsage* is the cores per seconds utilization of the user or account. Succinctly, this algorithm does a depth-first traversal of the tree of users and accounts, ordering decreasingly the users or accounts of each account by the quotient of their *RawShare* and *RawUsage*. It then evaluates if it is an account or a user. If it is an account, it does a recursive call on it. If it is a user, it assigns the fair share and returns.

The fair share is assigned by taking into account the total number of users, $N$, and the position in which the user was evaluated, $i$. The computation is just $\frac{N-i+1}{N}$, which means that the lower bound, that is, the worst fair share possible, is $\frac{1}{N}$, when $i = N$.

This method is meant to have shared accountability of the users for their entitlement to the machine. For example, if there are two sibling accounts, A and B, and A has a higher level fair share than B, then all users and accounts under A will have a higher fair share than B. So users are impacted not only by their own usage, but also the sum of their peers' usage.

## 2.3 Wrappers

In a shared HPC environment, queuing for resources is ever so frequent (Patel et al., 2020), and users have a limited impact on the priority of their jobs given the importance of fair share.

To reduce the time-to-solution of an Earth System Model (ESM) simulation workflow, the Autosubmit developers came up with a technique called task aggregation or wrapping. Their idea was to increase throughput by avoiding queuing subsequent tasks. For this reason they implemented vertical wrappers, which append workflow tasks into a longer submission.

In addition to vertical wrappers, horizontal wrappers were developed to comply with the platform's policy regarding the maximum number of jobs in the queue.

Finally, there is also the combination of the two types: vertical-horizontal and horizontal-vertical. A vertical-horizontal is made of multiple vertical wrappers running concurrently. Similarly, the horizontal-vertical is a single job made of multiple subsequent horizontal wrappers.

In all wrapper types, the dependencies among the tasks are respected and the underlying application task is not altered by their employment. Tasks are just submitted together to the remote platform. Therefore, all steps normally performed, such as saving the restart conditions (or checkpointing), are still executed. Moreover, if a task fails within the aggregated job, Autosubmit will relaunch the failed task without the need of a new job submission.

In this work, we will focus on vertical wrappers, as they are the proposed solution for the long queue times.

## 3  Methods

In this section, we explain the methodology employed in this study. First, in subsection 3.1, we provide a description of the HPC platforms we simulate and the hardware we used to run the simulations. Then we give an overview of the simulator in subsection 3.2. We cover the scheduling policy utilized in our experimentation in subsection 3.3. In subsection 3.4, we explain
how we carried out simulations with synthetic static workloads modeled after the (LUMI, 2024) supercomputer to address the modern job demands. In the subsection 3.5, we ran a dynamic simulation utilizing a real workload from a long decommissioned system to have a representative daily usage pattern. Finally, for modeling the workflow, we use the allocated CPUs and runtime of the job running the Nonhydrostatic Multiscale Model on the B-grid (NMMB) within the MONARCH application (Klose et al., 2021).

We perform a two-fold experimentation because we wanted to capture both a representative daily usage pattern (arrival times) and modern HPC job demands (requested CPUs, wallclock, user, and group identification).

### 3.1  HPC Platforms and Execution Framework

We wanted to model a large, shared, and general-purpose scientific machine. For that, we gathered data from the Large Unified Modern Infrastructure (LUMI) supercomputer (LUMI, 2024) operated by the EuroHPC and the Finnish Center for Science
(CSC). This is a flagship, modern, highly utilized, and scientific platform, within the top five machines according to the TOP500 list (Strohmaier et al., 2023). Also, given the sheer quantity of HPC resources, it is the center pillar for many European projects, such as Destination Earth (Hoffmann et al., 2023). We used this data to build synthetic workloads because we are confident that the job geometry (allocated CPUs and runtime) is representative.

For the dynamic workloads, we utilized the workload from the decommissioned Curie machine, which was operated by the
*Commissariat a l'Energie Atomic*, available in Feitelson and Tsafrir (2019) online repository.

Regarding the execution framework of this work, all simulations were carried out using our laptop computer, with an Intel i5-1135G7 and 16GB of RAM.

### 3.2  Slurm Simulator

We employed the BSC Slurm simulator (Ana Jokanovic, Marco D'Amico, and Julita Corbalan, 2018) to recreate the scheduler
behavior. This simulator is made of the same executables that make up Slurm, with only minimal changes to control the pace of time and the input data.

We implemented a prologue to the original run script (G. Marciani, 2024b) to use Slurm's account manager tool, *sacctmgr*, to include users and accounts, which were determined from the input workload file and configured before running the simulation. All users and accounts were given the same entitlement to the machine, in other words, the same *RawShares* value.

| Option | Value |
|---|---|
| *default_queue_depth* | 10,000 *jobs* |
| *defer* | True |
| *sched_interval* | 60 *s* |
| *bf_interval* | 60 *s* |
| *bf_max_time* | 30 *s* |
| *bf_resolution* | 1,800 *s* |
| *bf_window* | 10,080 *min* |
| *bf_continue* | True |
| *PriorityMaxAge* | 10 *days* |

**Table 1.** Main and backfill scheduler configuration (SchedMD, 2022). Configurations starting with *bf* apply to the backfill algorithm. These are used in MareNostrum 4, and we use them in all of our experiments.

Since the usage impacts the fair share value (SchedMD, 2019), it was important for us to ensure that all users have nil registered before the simulation. So we developed a Docker (Merkel, 2014) image (G. Marciani, 2024c) that ensures that the environment is clean, on top of being lightweight compared with using a virtual machine.

Finally, the simulator outputs a list of all the jobs that it processed, with the submit, start, end, and priority upon finish.

### 3.3 Slurm Configuration

In all of our experiments, we used MareNostrum 4's (BSC-CNS, 2023) scheduling configuration and priority weights, which are included in our repository (G. Marciani, 2024b). The scheduling configurations are specified in Table 1, where those configurations starting with *bf* apply to the backfill algorithm.

This policy uses the *multifactors* policy (SchedMD, 2023), where the fair share and age factor weights are $100,000$ and the size factor is $10,000$. The fair share factor is computed with the *Fair Tree* algorithm and with *backfill* as the secondary scheduler. We only adapted the total number of physical cores available for each platform we model: 360,448 for LUMI and 93,312 for CEA-Curie.

We chose this system's policy because it is from a large and general-purpose machine, besides it provided HPC resources to many groups across Spain and Europe, making it a highly demanded system.

### 3.4 Static Workloads

Static workloads are those where all jobs in the description are submitted at the same time. We made this decision to simplify the modeling, since we disregard the complex modeling of the arrival time (Cirne and Berman, 2000). We chose the number of jobs to be generated so that we stress the system but still within plausible bounds. This methodology was also employed by Jeannot et al. (2023) to recreate a HPC environment for performance analysis of their IO optimization technique.

| Statistic | CPUs | Runtime ($s$) | $CPU \cdot s$ |
|---|---|---|---|
| Mean | 774.2 | 3,952.7 | 3.4207e+06 |
| Std | 4,493.4 | 19,617.9 | 1.3920e+08 |
| Min | 1 | 0 | 0 |
| 25% | 48 | 23 | 3.5330e+03 |
| 50% | 256 | 125 | 3.9424e+04 |
| 75% | 1,024 | 902 | 2.1504e+05 |
| Max | 325,120 | 4,604,100 | 9.5056e+10 |

**Table 2.** Number of allocated CPUs, runtime, and core seconds statistics on the dataset captured in LUMI. 25% refers to the first quartile, 50%, the median, and 75%, the third quartile.

With this workload, we are unable to send workflow tasks with dependencies because of its single-submission design. Instead, we are interested in understanding how the different factors from the scheduling play a role in the queue time. So we take a single workflow task, and we vary its request, that is, its allocated CPUs, runtime, and the user's fair share. Then we track its wait time in the queue.

In this section, we explain how we analyzed and fit distributions to data from the LUMI supercomputer (LUMI, 2024), in subsection 3.4.1, and we describe how we set up the experimentation in subsection 3.4.2.

### 3.4.1 Workload Generator

We generated workloads by fitting a log-normal distribution to both the runtime and the allocated CPUs to the observed distribution using a script we developed G. Marciani (2024e). We achieved the following fitted values, in SciPy's nomenclature, for the allocated CPUs: $loc = -9.8930e - 1$, $scale = 1.9930e + 2$, and $s = 1.7195$; and, for the runtime: $loc = 9.5007e - 1$, $scale = 1.6506e + 2$, and $s = 2.6766$; with a sum squared error of, $1.5213e - 07$ and $1.4533e - 09$ respectively. In Figures 1 and 2, we have respectively the log allocated CPUs and log runtime cumulative distribution of the observed and generated data, in orange dashed line and blue solid, respectively.

Because the log normal distribution is unbounded, to generate the allocated CPUs we need to truncate it. Finally, for both the runtime and allocated CPUs the generated values were rounded to the nearest lower integer.

Once we generated the job's request, that is its runtime and allocated CPUs, we assigned it to one of the hundred users uniformly at random. We did the same for assigning users to accounts. If a task was attributed to a user without an account, we assigned it to one out of a hundred uniformly at random. We generated ten workloads to account for the variability of the job's characteristics. And we drew 1,000 jobs for each workload because it provided a congested yet plausible scenario.

### 3.4.2 Experimental Design

We added a single workflow task to the ten generated synthetic static workloads in order to track its queue time. With the goal of measuring the impact of the different wrapper configurations, we made the request of this single task be a multiple of

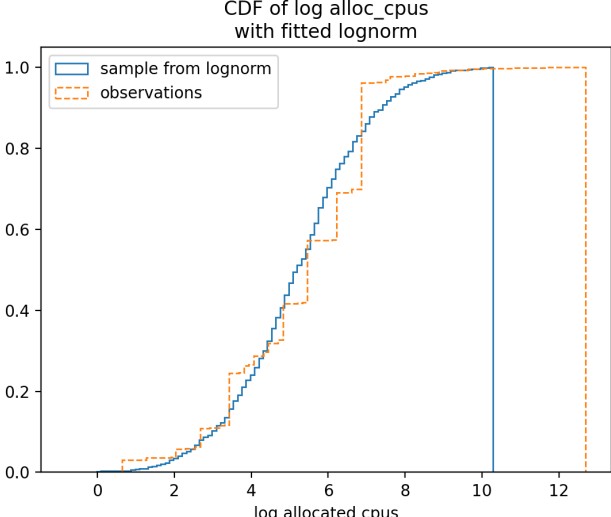

**Figure 1.** Cumulative distribution function of the log normal distribution for the allocated CPUs at LUMI. The orange dashed line is the log observed cumulative allocated CPUs and the blue solid line is a random sample generated with the parameters $loc = -9.8930e - 1$, $scale = 1.9930e + 2$, and $s = 1.7195$.

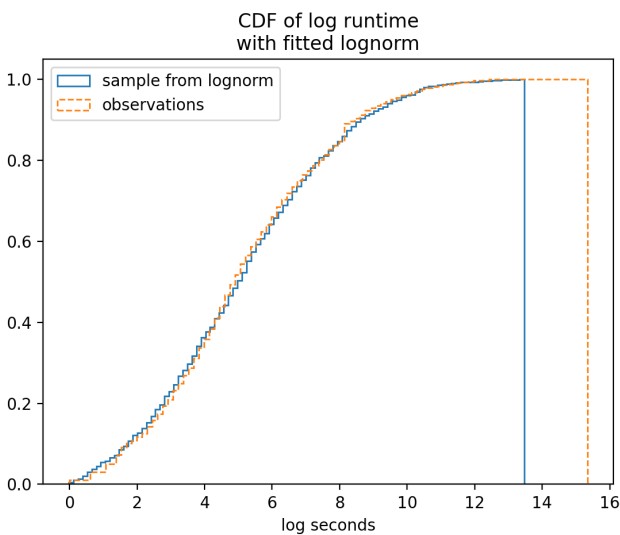

**Figure 2.** Cumulative distribution function of the log normal distribution for the runtime at LUMI. The orange dashed line is the observed cumulative log runtime of jobs, with runtime different than 0, and the blue solid line is the cumulative distribution of a random sample generated with the parameters $loc = 9.5007e - 1$, $scale = 1.6506e + 2$, and $s = 2.6766$.

1,800 seconds of runtime and 96 CPUs. We take these values from executions in MareNostrum 4 of the NMMB model of the MONARCH application (Klose et al., 2021), simulating a single day in the reanalysis configuration.

In order to control the fair share, given that all the jobs of the workload are submitted at the same time, we preceded the simulation with a batch of "dummy" jobs so that all users have usage recorded. Otherwise, all users would have nil usage, therefore maximum fair share, and it would effectively remove the fair share from the scheduling. We set all the users "dummy" submission runtime to be proportional to the synthetically generated usage, except for the user employed with launching the workflow, for whom we chose its usage to control its fair share. This was done because we know that users have a recurrent pattern of utilization, as described by Patel et al. (2020). Therefore, the introduction of dummy jobs made the fair share of the users coherent with respect to the synthetically generated usage.

We tested every combination of 1,800, 3,600, 7,200, and 12,600 seconds of runtime and 96, 192, 384, 672, and 1,152 CPUs. For the fair share of the user launching the workflow 2.2, we tested 0.1, 0.2, 0.25, 0.3, 0.5, 0.7, 0.75, 0.8, and 0.9.

### 3.5 Dynamic Workload

Opposed to the static workloads, jobs are submitted at different times in dynamic workloads. This subsection explains how we set up the experiment for them. In section 3.5.1 we cover the choice of workload, and in section 3.5.2 we cover how we set up the experiment.

#### 3.5.1 Workload Choice

We used the CEA-Curie clean version 2 from Feitelson and Tsafrir (2019) repository, which only considers those jobs submitted after a major upgrade in the machine and removes those which were certainly badly logged, i.e., reported running for far too long. This dataset is one of the largest publicly available workloads for a machine following the criteria we want: a large general purpose shared scientific machine.

We explored the workload in search of periods of congestion, which accumulate normally on Thursdays and Fridays on this machine and last a few days. We selected one week of the trace, between 9/6/2012 and 15/6/12. The resulting workload file is found at the code snippet alongside with the script to add the workflow (G. Marciani, 2024a).

#### 3.5.2 Experimental Design

To test if wrappers improve the time-to-solution, we considered a seven chunk split execution of the MONARCH application, with the tasks requesting 96 CPUs and taking 1,800 seconds to simulate a day in the reanalysis configuration, typical values of its execution in MareNostrum 4.

The simulator does not have support for dynamic submission times, i.e., only submit a constrained job when its dependency has finished, as it would be the case in real life with Autosubmit. Therefore, we had to specify the submission time on the workload file, prior to knowing when the tasks will be scheduled, and consequently finish.

| Label | Submission Instant |
|:-----:|:------------------:|
| A.1 | 14/6/2012 at 10 |
| A.2 | 14/6/2012 at 15 |
| A.3 | 14/6/2012 at 20 |
| B.1 | 15/6/2012 at 10 |
| B.2 | 15/6/2012 at 15 |
| B.3 | 15/6/2012 at 20 |

**Table 3.** Submission instants of the workflow and their corresponding label.

Therefore, to minimize the waiting time of a constrained task from adding too much queue time, increasing its age factor, we defined its submission time as the instant its predecessor would have finished if it did not have any waiting time. That is, the second task is submitted 1,800 seconds after the first one; the third, 3,600 seconds after the first one; and so on and so forth.

In order to control the fair share of the user executing the workflow, we measured the usage up until the submission instant of all users by executing the workload with no MONARCH tasks added to it. With this utilization, and taking into consideration that all accounts have the same entitlement to the resources, the fair share will be roughly one minus the percentile of the distribution of usage of the accounts. For instance, to have a fair share of 0.2 we need to match the utilization of the 80th percentile of the usage of the accounts.

We tested multiple fair share values: the worst, in this case is 0.01 – given the number of users in the machine 2.2 –, 0.2, 0.3, 0.4, 0.5, 0.6, 0.7, and the best, which is 1.0.

Then we considered the unwrapped case, in which all the tasks are launched individually, and the wrapped case, where we created a single job adding the runtime of all of them. In the case, that meant a job of length 12,600 seconds and 96 CPUs.

We simulated both unwrapped and wrapped cases, at six different submission times: at 10, 15 and 20 of both Thursday 14th and Friday 15th. We did this to account for the variability in the utilization by the users because it is unknown and, therefore, random.

In Table 3, we label these submission instants and plot them in Figure 3 as black vertical dashed lines along with the evolution of in-queue and in-use resources in orange dashed and in blue solid lines, respectively, from a simulation of the original trace with no workflow added to it. The dashed blue line is the total number of CPUs of the CEA-Curie machine. We observe two clear orange dashed peaks indicating the daily usage cycle and the blue solid line with a clear maximum, which is the total available CPUs. In all submission instants we observe jobs being queued.

## 4 Results

In this section, we analyze the results of the runs of both types of experiments: the static workloads in subsection 4.1 and the dynamic workload in subsection 4.2.

Full results tables were omitted due to space constraints, but they are available in the Zenodo platform G. Marciani (2024f).

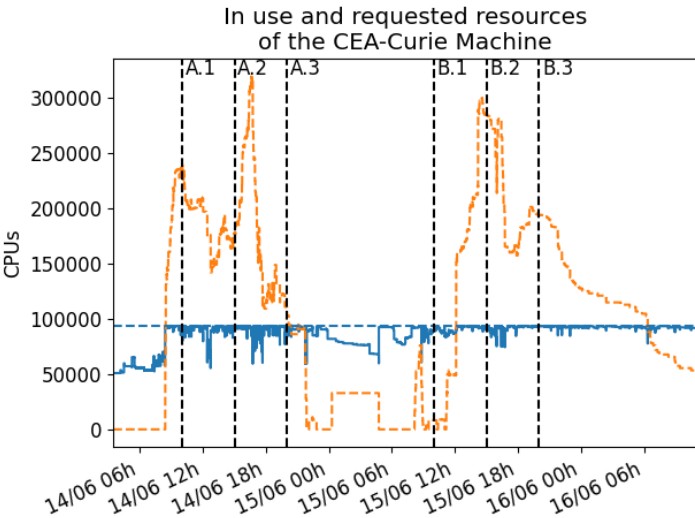

**Figure 3.** Simulated usage, in blue solid, and in queue resources, in orange dashed lines, from the untouched CEA-Curie workload for the week from 9/6/2012 to 15/6/12. The vertical black dashed line indicates the instant of submissions with labels following Table 3 and the blue dashed horizontal line is the total number of CPUs of the machine.

## 4.1 Static Workloads

We compute the correlation of the runtime, allocated CPUs, and fair share with the average, maximum, and minimum of both the queue time and the priority upon finishing the job, respectively $T_q$ and $P$, in Table 4. We observe that the fair share is the dominant factor on the waiting time, with a clear inverse correlation of $-0.87$ between the average wait time and fair share. Moreover, we observe the independence of the allocated CPUs and runtime with respect to the queue time.

Additionally, in Figure 4 we plot the average wait time across all ten experiments for every job configuration in function of the fair share factor, where we see an exponential reduction in queue time as we increase the fair share, but we observe that its impact is smaller for the lower fair share values, visible from the spread of the each of the lines representing a job configuration. After all job configuration's merge into one. We fit an exponential model (G. Marciani, 2024h), where $a$ is the exponent constant and $b$ is the multiplicative one.

As for the priority upon finishing, we observe in Table 4 how the fair share is almost completely positively (0.99) correlated with the average, maximum, and minimum priority. The rest of the factors, allocated CPUs and runtime, show no correlation.

Analogous to the wait time, we plot in Figure 5 the average across all ten experiments of the priority in function of the fair share, for each and every combination of allocated CPUs and runtime. There, we see how the priority grows linearly with respect to the fair share, with the largest deviations in the low fair share spectrum.

|            | Runtime  | CPUs    | Fair share |
|------------|----------|---------|------------|
| $max\ T_q$ | 0.0229   | -0.0161 | -0.8379    |
| $min\ T_q$ | 0        | 0       | -0.8572    |
| $\overline{T_q}$ | 0.0048 | 0.0007 | -0.8720 |
| $max\ P$   | 0.0018   | 0.00623 | 0.9911     |
| $min\ P$   | -0.0002  | 0.00026 | 0.9929     |
| $\overline{P}$ | -0.0010 | 0.00128 | 0.9952  |

**Table 4.** Correlation table of the job and fair share with queue time and priority. $T_q$ is the time in queue and $P$ is the priority upon finishing of the job. An overline indicates the average across the ten experiments and $max$ and $min$ are the maximum and minimum observed across the 10 experiments of the particular quantity.

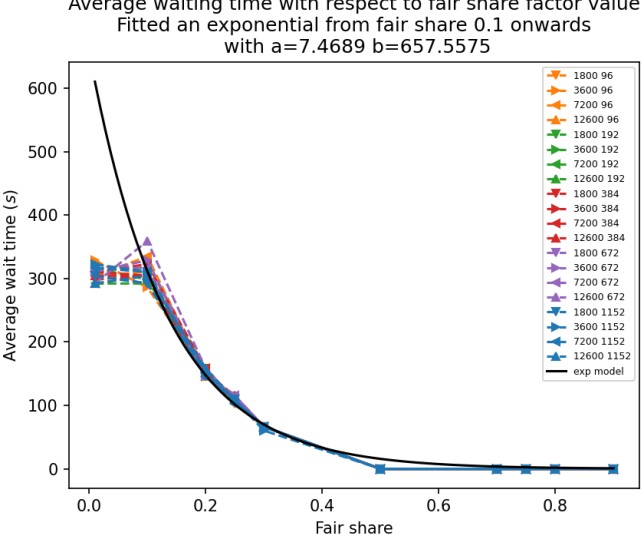

**Figure 4.** Average of the queue time across all experiments with respect to the fair share value. Each color represents a different runtime from 1,800, 3,600, 7,200, and 12,600 seconds, and each line style represents a different allocated CPUs configuration from 96, 192, 384, 672, and 1,152.

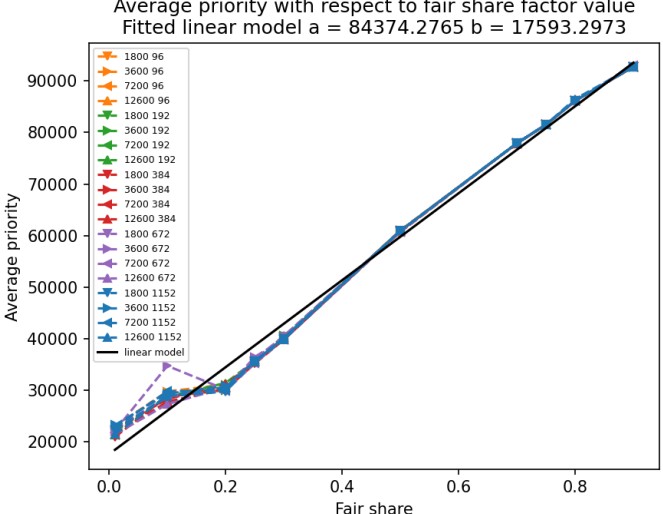

**Figure 5.** Average of the priority time across all experiments with respect to the fair share value. Each color represents a different runtime from 1,800, 3,600, 7,200, and 12,600 seconds, and each line style represents a different allocated CPUs configuration from 96, 192, 384, 672, and 1,152.

## 4.2 Dynamic Workload

We gathered all six instants and made a box plot of the difference of the time-to-solution (i.e., the difference between submission time and end of the workflow) between the unwrapped and the wrapped executions. Thus, a positive value indicates that the wrapped workflow tracked less queue time than the unwrapped counterpart.

In Figure 6, we see that the difference is positive on average, indicated by the green triangle. Also, the median, indicated by the orange line, is also always positive. However, we observe fliers where the unwrapped outperformed the wrapper. This is the case for the worst fair share, 0.4, and 0.6.

## 5 Discussion

As seen in Figure 6, we achieved a reduction in queue time across all fair share values in the dynamic results. On average, this reduction was 1%, reaching up to a 7% decrease in queue time relative to the total workflow runtime. These results support the hypothesis that the reduction is caused by avoiding multiple submissions.

Since we observed consistent (on average and on median) reductions when using aggregation, we anticipate greater gains in longer workflows running in congested environments because bigger wrappers can be created, reducing the number of submissions and therefore the time spent waiting for resources.

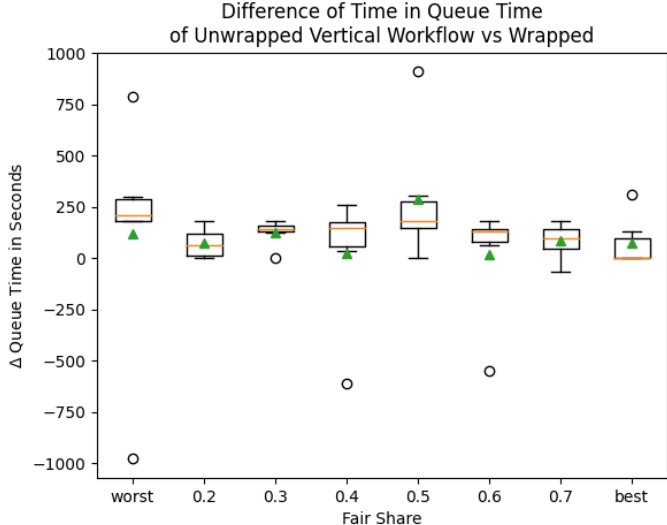

**Figure 6.** Box plot of the difference of the unwrapped queue time with respect to the wrapped. Green triangle indicates mean, orange horizontal line indicates the median. Circles indicate fliers, which are instants of submissions that surpass 1.5 times the interquartile difference.

Additionally, the 7% figure could be greater if we consider that the machine had only two days of congestion per week. Current flagship systems are usually congested, and it is not uncommon for jobs to queue for days.

We observed three negative outliers, that is, the instances where the unwrapped workflow stood less in queue than the wrapped. We believe those were particular instances where the combination of running jobs and queuing jobs allowed the backfill algorithm to schedule the separate tasks earlier because they were smaller. Under this scenario, as we increase the length of the job, it is less likely that the scheduler would have free spots to execute, so the job has to wait to be scheduled by its priority. But, given the few instances where we observed arrangement, we believe this scenario is unlikely.

As for the static results, where we captured modern job requests (i.e., the requested CPUs and wallclock) modeled after the LUMI supercomputer, these give us an idea of what are the meaningful factors that immediately impact after submission under a stressed environment. What we see is a total correlation of the queue time with respect to the fair share, and, even stronger, an exponential relationship. This means that with a fair share of just 0.5, it is enough to have more priority than all the other jobs in the queue, and therefore to be scheduled immediately.

This exponential relation comes from the way we set the fair share to all the synthetically generated users. Since their prior usage was set to be proportional to the generated workload, their fair share followed the log-normal distribution.

Regarding job requests in static executions, we observe in Table 4 that both runtime and allocated CPUs are independent of queue time. This is explained because these simulations are short compared with the dynamic ones, not allowing for the age factor to build up, and neither the backfill algorithm to be noticeable.

Finally, Figure 5 shows the expected linear relationship between priority and fair share. This is due to the weighted sum that Slurm uses to compute the priority in Equation 4. The slope of the fitted line, 84,374, nearly matches the fair share weight of the Slurm configuration that we used, which is 100,000.

As is the case for the queue time, the requested CPUs and runtime are not correlated with the priority. This is due to the aforementioned shortness of the simulations and the relatively small application request with respect to the large system. Taking the maximum time that any of our tracked jobs was in the queue, 372 seconds, adds just 61 (372 divided by 7 days in seconds multiplied by $10^5$) to the priority from the age factor. And if we do the same for our largest request in CPUs, 1,152, it adds just 32 (1,152 divided by the total available CPUs in LUMI, 360,448, multiplied by $10^4$) to the priority due to the size factor. Compared with a fair share of just 0.01 that adds 1000 (0.01 times $10^5$) to the priority, both the age and the size factors are marginal.

## 5.1 Limitations

In this subsection, we discuss the major weaknesses that we identified during our work.

First, the Slurm simulator does not support dynamic submissions (i.e., launching a job the moment its dependency finishes). Therefore, we had to define the submission time in advance by assuming the best-case scenario, in which no task is delayed. Thus, the age factor increases while the job waits in the queue for its dependency to finish, resulting in a higher priority than they would have in reality.

However, with the configuration we tested, we found that the priority added by the age factor was marginal. The maximum time that a job was in the queue in any of our simulations was 1,776 seconds, in the worst fair share case submitted at 15/6/2012 at 20 (submission instant B.3). This adds just 293 (1,176 seconds divided by the total number of seconds in seven days multiplied by $10^5$) to the job's priority. This is minimal compared with the priority added by a fair share of just 0.01, which, after being multiplied by its corresponding weight of $10^5$, adds 1,000 to the job's priority.

Another limitation is that the simulator does not support node sharing among jobs, as is the case in MareNostrum 4. Therefore less than a node requests would be scheduled to whole nodes, whereas, in MareNostrum 4, they would share resources. But, we have seen systems enforce node exclusivity across the board, as is the case with LUMI.

Finally, the Slurm simulator here employed is not determinist, although the authors of the BSC contribution to it greatly reduced it (Ana Jokanovic, Marco D'Amico, and Julita Corbalan, 2018). This is another reason to run multiple experiments and take the average.

## 6 Conclusions

The time in queue of the simulations is a growing issue for those users utilizing highly congested machines. And it is even worse for those executing simulations with vertical workflows, where workflow tasks have to wait until their dependency is met. This paper analyzes for the first time, to the best of our knowledge, a simple but powerful solution to mitigate this issue, which is to aggregate tasks into a single submission to be sent to the HPC platform.

We measured the impact of aggregating tasks by running two simulations under the same conditions with a Slurm simulator, one wrapping and another with all the tasks independently submitted. We used the MONARCH air quality application as a reference for the application. To have both modern job requests and realistic behavior on the usage of the machines, we performed two experiment types: one with synthetic static workloads modeled after a current flagship system, where all the jobs are submitted at the same time, and another with a real dynamic workload, as recorded in a production machine.

We tied this impact with a key scheduling factor employed by system administrators of Slurm: the fair share. This factor quantifies the responsiveness of the machine to the user, and is shared with the group. This factor is normally assigned a high weight, which equates to a larger impact in the job's priority.

Our findings support the thesis that aggregating subsequent tasks reduces queue time. We observed a maximum difference in queue time between unwrapped and wrapped of 7% of the total runtime of the workflow. If we put this saving in terms of the 10% to 20% queue overhead reported by the community, that would be a reduction of the queue time of 35% to 70% percent.

This result supports our thesis that, since we submit fewer jobs, we have fewer tasks of the workflow stuck in the queue, and therefore the overall time-to-solution of the simulation is improved.

As for the fair share, we see how a fair share factor of just 0.5 is enough for the job to be executed immediately in the static experiments. This is not as clear in the dynamic results, but we observe that with a higher fair share, the impact of using wrappers decreases.

Finally, this paper brings to the forefront the issue of queue time in shared HPC platforms, which is particularly acute for the climate community, but it may be shared with other communities running their applications on congested shared HPC machines. We expose the important parts of the inner workings of the scheduling algorithm of the most popular workload manager, Slurm, and relate them to the request in terms of CPUs and wallclock of the workflow. This work is of interest to all those who see the need to optimize their time-to-solution by providing recommendations on how to submit their tasks and the rationale as to why it is beneficial.

*Code and data availability.* The Slurm simulator code is available in https://earth.bsc.es/gitlab/mgimenez/ces_slurm_simulator under GPL2 license. The exact version of the simulator used to produce the results in this paper is archived on Zenodo (G. Marciani, 2024b).

The Docker image to run the Slurm simulator is available in https://earth.bsc.es/gitlab/mgimenez/docker-ubuntu-ces-slurm-sim under GPLv3 license. The exact version used in this paper is archived on Zenodo (G. Marciani, 2024c).

Analysis scripts for the workload and results are available, respectively, in https://earth.bsc.es/gitlab/mgimenez/scripts-hairball/-/snippets/128 and https://earth.bsc.es/gitlab/mgimenez/scripts-hairball/-/snippets/131 under GPLv2. The exact version used in this paper is available on Zenodo, respectively, G. Marciani (2024e) and (G. Marciani, 2024h).

Script to include workflow to Curie dynamic trace is available in https://earth.bsc.es/gitlab/mgimenez/scripts-hairball/-/snippets/125. The exact version used in this paper is available on Zenodo under GPLv2 (G. Marciani, 2024a).

The results of the simulations are made available in the Zenodo platform G. Marciani (2024f).

The original data and the input workload files, for both static and dynamic simulation, are made available on the Zenodo platform, respectively, G. Marciani (2024g) and G. Marciani (2024d).

Due to its sensitivity, data gathered from the LUMI supercomputer is not publicly available.

*Author contributions.* MGM has developed the methodology, the Docker image of the BSC Slurm simulator, the Python library for managing the Standard Workload Manager, ran the simulations, performed the analysis, and wrote the paper. MC has conceptualized the work, supervised the work, and revised the manuscript. GU and MCA have supervised the work and reviewed the manuscript. BPK he has reviewed the manuscript. FDR has provided funds for the development of the work.

*Competing interests.* The authors declare that they have no conflict of interest.

*Acknowledgements.* This work received support from grant CEX2021-001148-S-20-5 funded by MICIU/AEI/10.13039/501100011033 and by FSE+. Mario Acosta is supported from the Spanish National Research Council through OEMES (PID2020-116324RA-I00). The authors declare they used DeepL in order to improve readability of the manuscript in all sections. After using this tool, the authors reviewed and edited the content as needed and take full responsibility for the content of the published article.

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
