# Peer review of "Evaluating the Impact of Task Aggregation in Workflows with Shared Resource Environments: use case for the MONARCH application"

_EGUsphere, 2025_

## Author Comment (AC5)

We would like to thank the reviewer for their constructive comments. We will address each point raised by the reviewer in the original order and numbering. We have highlighted the reviewer's comments in yellow. Changes made to the manuscript in response are included in blue highlights.

**Major points:**

1. The material is not put well in the context of prior work, not only on measurement (e.g. Balazji et al 2017, whose results differ from those presented here - my quick calculation from their table 2 suggests the problem is of order 30%, so I don't know where the 10-20% they use as their motivation came from), and the large literature on pilot jobs on distributed and high throughput computing. The concept of wrapping things together is quite well established, but that leads me to my second concern:

We thank the reviewer for comment. We agree that the figure representing the overhead due to queue time was unclear. We have therefore changed the introduction (lines 30–35) to clarify it. It now reads:

However, lately the community has drawn attention to the efficiency of the simulations, taking into account the most demanding part along with the postprocessing, failure, and time spent in the queue. With this in mind, Balaji et al. 2017 proposed a set of performance metrics for Earth system model simulations. Among these metrics, the authors proposed the simulated years per day (SYPD), which is the ratio of the time simulated in years with respect to the runtime of the job in days, as well as the actual simulated years per day (ASYPD), which is the simulated time in years divided by the the time-to-solution of the simulation. Thus, this metric now accounts for time in the queue and also system interruptions.

In Acosta et al. 2024, the authors computed these metrics for 33 CMIP6 simulations executed on 14 machines. Their analysis showed that the difference between ASYPD and SYPD ranged from 0% to 78%. But, they noted that not all institutions reported ASYPD consistently. Some accounted for both interruptions and queue time, while others accounted only for queue time. For those institutions that only accounted for queue time, the spread was between 10% and 20%. The authors therefore concluded ``that queuing time represents an increment of around 10%–20% of the speed of the ESM.''

Regarding the contextualization of the work, we agree that pilot-job systems share a fundamental characteristic with wrappers, making them an essential topic for the introduction. Moreover, this community would also benefit of the results and discussions of our work.

In addition to the pilot-job systems, we have expanded the introduction to include other workflow managers that offer solutions similar to those of Autosubmit's wrappers. Lines 53-59 have been altered and now read:

Aggregation was implemented in other fields, with different degrees of sophistication. In Earth Sciences, Mickelson et al. 2020 suggested using Cylc's feature to submit multiple jobs to reduce the queue time. Both Aiida (Huber et al. 2020) and Snakemake (Mölder et al. 2021)— from the material and life sciences fields, respectively — provide a way of to submit multiple workflow tasks as a single job. The former implements this via a ``metascheduler'' plugin (HyperQueue plugin), while the latter refers to aggregation as ``grouping.''

Moreover, Pilot-job systems were developed to increase workflow throughput, but later evolved into more sophisticated solutions (Turilli et al. 2018). These systems are characterized by implementing "resource placeholders, multi-level scheduling, and coordination patterns to enable task-level distribution and parallelism on multi-tenant resources." One major example of a modern pilot-job system is Radical-PILOT (Merzky et al., 2021)

Wrappers share some of the features of a typical pilot-job system. Wrappers provide a simpler ``resource placeholder'', where all the task requests are added into a large submission. However, their objective is to either increase throughput by reducing submissions or to comply with maximum job restrictions. Therefore, besides fault tolerance, wrappers do not improve task scheduling and coordination within the allocation.

2. One of the reasons that this is not done substantially in large-scale climate simulation is that the "chunks" of simulation (their nomenclature) provide natural checkpointing, and with bigger jobs than those they discuss model and hardware failures do lead to the need to checkpoint. Hardware failures are more prevalent now, and with bigger jobs that can be problematic. Perhaps this is why they only did Vertical Wrapping (they are only using one node per job if they only need 96 cores). Possibly horizontal wrapping would be more subject to this issue, but it might also be possible for the workflow manager to cope.

We thank the reviewer for the comment on checkpointing. We have expanded the introduction to clarify that the wrappers neither remove nor alter the frequency of the application's checkpoints.

Additionally, we explained the reasoning behind each type of wrapper (vertical and horizontal). We have revised the manuscript in the "Wrapper" subsection of the "Background" section (lines 116-126), that now reads:

In a shared HPC environment, queuing for resources is ever so frequent (Patel et al., 2020), and users have a limited impact on the priority of their jobs given the importance of fair share.

To reduce the time-to-solution of an Earth System Model (ESM) simulation workflow, the Autosubmit developers came up with a technique called task aggregation or wrapping. Their idea was to increase throughput by avoiding queuing subsequent tasks. For this

reason they implemented vertical wrappers, which append workflow tasks into a longer submission.

In addition to vertical wrappers, horizontal wrappers were developed to comply with the platform's policy regarding the maximum number of jobs in the queue.

Finally, there is also the combination of the two types: vertical-horizontal and horizontal-vertical. A vertical-horizontal is made of multiple vertical wrappers running concurrently. Similarly, the horizontal-vertical is a single job made of multiple subsequent horizontal wrappers.

In all wrapper types, the dependencies among the tasks are respected and the underlying application task is not altered by their employment. Tasks are submitted together to the remote platform. Therefore, all steps normally performed, such as saving the restart conditions (or checkpointing), are still executed. Moreover, if a task fails within the aggregated job, Autosubmit will relaunch the failed task without the need of a new job submission.

In this work, we will focus on vertical wrappers, as they are the proposed solution for the long queue times.

In this work, we will focus on vertical wrappers, as they are the proposed solution for long queue times.

We rewritten the abstract to clarify that wrappers do not alter the underlying application (lines 2–19). The new text reads as follows:

High Performance Computing (HPC) is commonly employed to run high-impact Earth System Model (ESM) simulations, such as those for climate change. However, running workflows of ESM simulations on cutting-edge platforms can take a long time due to the congestion of the system and the lack of coordination between current HPC schedulers and workflow manager systems (WfMS). The Earth Sciences community has estimated the time in queue to be between 10% to 20% of the runtime in climate prediction experiments, the most time-consuming exercise. To address this issue, the developers of Autosubmit, a WfMS tailored for climate and air quality sciences, have developed wrappers to submit multiple subsequent workflow tasks -- the atomic unit of compute in the workflow -- as single submission, without changing them. However, although wrappers are widely used in production for community models such as EC-Earth3, MONARCH, and Destination Earth simulations, to our knowledge, the benefits and potential drawbacks have never been rigorously evaluated. Later, the developers of Autosubmit noticed that the performance of the wrappers was related to the past utilization of the user which reflects on job priority in Slurm via the fair share factor. The objective of this paper is to quantify the impact of wrapping subsequent tasks on queue time and understand its relationship with the fair share and the job's CPU and runtime request. To do this, we used a Slurm simulator to

reproduce the behavior of the scheduler and, to recreate a representative usage of an HPC platform, we generated synthetic static workloads from data of the LUMI supercomputer and a dynamic workload from a past flagship HPC platform. As an example, we introduced jobs modeled after the MONARCH air quality application in these workloads, and we tracked their queue time. We found that, by simply joining tasks, the total time-to-solution of the simulation reduces up to 7% with respect to the runtime of the simulations, and we believe that this value is larger the longer the workflow. This saving translates to absolute terms of about eight days less wasted in queue time for half of the simulations from the IS-ENES3 consortium of CMIP6 simulations. We also identified a high inverse correlation of -0.87, between the queue time and the fair share factor in the static experiments.

We have also changed the introduction (lines 48-51) to further emphasize that using wrappers does not affect the application.

Observing the impact of fair share on the priority, the developers of Autosubmit (Manubens et al., 2016), a Workflow Manager System (WfMS) designed for climate and air-quality applications, have implemented a task aggregation technique called ``wrappers.'' The idea is that Autosubmit aggregates, or wraps, multiple subsequent tasks into a single job submission to save the users from queuing multiple jobs under a low priority without changing the underlying script that executes the task. Additionally, wrapping is also employed for concurrent tasks in order to comply with the maximum number of jobs in the queue.

3. The use of MONARCH and 96 core jobs (which are small for these systems) means that these results might not be typical of bigger jobs, not least because most large systems also prioritise bigger jobs, and so the fair-share factor influence may be less important. There is a large problem space that has not been examined. It's certainly the case that the CMIP extrapolation cannot be substantiated without exploring this in their simulations.

We thank the reviewer for the comment. We acknowledge that the jobs used in our experiments (96 cores) are small compared to typical CMIP simulations. And it is absolutely true that bigger jobs, in terms of cores, normally have more priority from the size factor. But our subject of study are the vertical wrappers, which were developed to avoid requeuing. Therefore we have clarified our hypothesis in lines 62-64 and now it reads:

Although aggregation is utilized in other fields and also by Autosubmit users, with positive impact reported, there is a lack of understanding of the reasons and conditions under which aggregating reduces queue time. Therefore, in this work, we tested whether wrapping subsequent tasks together reduces queue time and if the fair share is the most important factor in reducing queue time.

Moreover, following on the reviewer's comment on the impact of size and fair share, we have extended the subsection "Scheduling Algorithms of Slurm" in the background section,

lines 85-94, to explain how each one of the factors of the priority are computed. Now it reads:

When a job is submitted, Slurm immediately attempts to allocate resources to it. If no resources are available, the job will be placed in a queue. This queue is sorted in descending order according to the priority integer, which updates periodically.

Slurm's scheduling design consists of two coordinated algorithms that traverse the ordered list of queued jobs and attempt to allocate resources to them. The main algorithm tries to schedule as many jobs as possible from the queue. If it fails because of a lack of resources, it breaks the loop and sleeps. There are two options for the second algorithm: built-in and backfill. The first option works like the main algorithm. The second option is the backfill algorithm (Srinivasan et al., 2002).

The backfill algorithm is an optimization technique that allows jobs with lower priority to be scheduled before higher-priority jobs, as long as they do not interfere with the start time of the higher-priority jobs. Therefore, the smaller the job is in terms of wallclock and/or CPUs requested, the more likely it is to be backfilled. Without considering the effects of the backfill algorithm, jobs with higher priority are scheduled earlier.

Slurm uses the multifactor policy by default to compute the priority integer. In Slurm terminology, factors are always floats stored in double-precision between zero and one. The priority is computed as a weighted sum of the following factors: age, size, association, fair share, and quality of service (QoS). A 32-bit encoded integer weight is associated with each of these values by the configuration.

The age factor, which is the one related to the time in queue, starts at zero and linearly grows until reaching the maximum, which is configured by setting the flag PriorityMaxAge in the Slurm configuration file. The default is seven days. In Equation 1 we have the expression used to compute the factor where $t$ is the time elapsed from submission, discounted by the time voluntarily held by user, and $T$ is the total amount of seconds set by the administrator with flag PriorityMaxAge.

$$a_i(t) = \begin{cases} \frac{t}{T}, & t \leq T \\ 1, & t \geq T \end{cases} \cdot (1)$$

By default, the size factor is the proportion of CPUs used by the job with respect to the total available. Therefore, when the whole machine is requested, the maximum is reached. System administrators can invert this by setting the flag PriorityFavorSmall}to True, which would give job requesting a single CPU the maximum value.

The size factor is computed as in Equation 2, where $r_i$ is the number of CPUs requested by job $i$ and $S$ is the total number of CPUs on the machine.

$$s_i = \frac{r_i}{S} \quad or \quad s_i = \frac{S - r_i + 1}{S}. (2)$$

QoS stands for quality of service. It is a mechanism that system administrators normally use to benefit smaller jobs (such as debug execution), since by default Slurm gives more priority to larger jobs. From the user side, this means higher priority upon submission, hence less waiting time, at the cost of a more constrained job submission. System administrators typically create different QoSs according to job length and size — for example, standard, debug, xlarge, and xlong.

Each QoS is configured with an associated priority, which is another integer value. Then the factor is computed as the proportion with respect to the largest priority of all QoSs. Therefore, if the job $i$ is submitted to QoS $u$ with priority $q_u$, its QoS factor is calculated according to Equation 3, where $Q$ is the set of the priorities of all QoSs configured by the system administrator.

$$q_i = \frac{q_u}{max_{q \in Q} q}. (3)$$

The fair share factor is the one that aims to balance the resources among users. It is computed in two different ways, which will be addressed in its separate Section 2.2.

Finally, the job's priority is computed according to Equation 4. $P_i(t)$ Is job $i$'s priority at time $t$. $a_i$ is the age factor, $w_a$ the age factor weight, $s_i$ is the size factor, $w_s$ the weight factor associated with size, $f_i$ is the fair share of the user that launched job $i$, $w_f$ is the weight, and finally $q_i$ and $w_q$ are the factor and weight associated to QoS.

$$P_i(t) = a_i(t) \cdot w_a + s_i \cdot w_s + f_i \cdot w_f + q_i w_q. (4)$$

Finally, given the weights that size and fair share factors are normally assigned, we have that to match an extremely low fair share it would be necessary to make an extremely large request in CPU resources.

For example, in MareNostrum 5 — and it also applied for the previous version of the machine — the highest weight is the given to the QoS factor (10^6), followed by age and fair share (10^5), and finally size (10^4). Taking the largest simulation CMIP6 simulation analyzed by Acosta et al. 2024 in terms of CPUs, NERC-HadGEM3-GC3.1-HH, which requested 12,024 of the 118,080 available CPUs in the Archer x30 system, the size factor adds to the job a priority of 1,018 (12,024/118,080*10^4 = 1,018). This is nearly matched by an extremely low fair share of 0.01 (0.01*10^5 = 1,000). In the case of Destination Earth, their 30 year simulation runs on 305 nodes of the MareNostrum 5's general partition. This means the size factor adds to the job a priority of 470 (35,160/725,760*10^5 = 470) , as 35,160 out of the 725,760 available cores are requested. The aforementioned fair share of 0.01 (0.01*10^5 = 1,000) more than doubles the priority that the size factor gives to the job.

This argument is fundamentally dependent on the configuration of the system. What we observe is that the majority of the EuroHPC systems have weights that make the fair share more impactful on the job's priority than the size. In Lumi, both have the same weight (1,000), on MareNostrum fair share has ten times more impact (100,000 for fair share and 10,000 to size), and Leonardo is the only system that places a thousand times more weight to the size (10,000,000) versus the fair share (25,000).

**Minor points:**

1. There is a language (ASYPD, SYPD) for the difference between the overall throughput and the peak throughput that was introduced in Balaji et al (2017) - a paper that shares a co-author with this one, so it is surprising that language is not used, and that Balaji et al is not cited

We appreciate the suggestion to include the ASYPD and SYPD metrics since they are the standard for measuring the performance of climate and weather simulations. However, despite their strengths, they have an important temporal component, i.e., the total time simulated, which does not translate to other applications. For example, aggregation could be emplyed in genome sequence alignment applications typically executed with Snakemake [1] or material sciences simulations using SIESTA [2] with AiiDa [3]. Therefore, we prefer to continue using the universal metrics of time-to-solution and queue time, which applies across all fields.

But, following on the reviewer's comment, we have included Balaji et al (2017) reference to the manuscript, where we introduce the issue of queue time and climate simulations (lines 30-35). It now reads:

However, lately the community has drawn attention to the efficiency of the simulations, taking into account the most demanding part along with the postprocessing, failure, and time spent in the queue. With this in mind, Balaji et al. 2017 proposed a set of performance metrics for Earth system model simulations. Among these metrics, the authors proposed the simulated years per day (SYPD), which is the ratio of the time simulated in years with respect to the runtime of the job in days, as well as the actual simulated years per day (ASYPD), which is the simulated time in years divided by the the time-to-solution of the simulation. Thus, this metric now accounts for time in the queue and also system interruptions.

In Acosta et al. 2024, the authors computed these metrics for 33 CMIP6 simulations executed on 14 machines. Their analysis showed that the difference between ASYPD and SYPD ranged from 0% to 78%. But, they noted that not all institutions reported ASYPD consistently. Some accounted for both interruptions and queue time, while others accounted only for queue time. For those institutions that only accounted for queue time, the spread was between 10% and 20%. The authors therefore concluded ``that queuing time represents an increment of around 10%–20% of the speed of the ESM.''

[1] https://workflowhub.eu/workflows/547

[2] https://doi.org/10.1063/5.0005077

[3] https://phantomsfoundation.com/AI4AM/2025/Abstracts/AI4AM2025_Garrido_Jaime_134.pdf

2. I had not seen Abhinit etal 2022, so I looked at it. I do not think it is saying the same thing as stated here. The problem in climate is unlikely to reach a need to wrap more than dozens of tasks (any more and the checkpoint issue dominates), whereas Abhinit et all were looking at wrapping thousands of tasks - in their case the issue is that most SLURM configurations do not have enough memory or resources to deal with the look ahead for queues with thousands of tasks. (Those that do are typically configured for High Throughput Computing, which is a different configuration to those encountered in most HPC sites where climate simulation is undertaken.)  Abhinit et al's discussion is more relevant to Dask workflows than simulation workflows. That said, it is indeed the case that HPC sites often limit the number of jobs users can have in queues, which is why tools like Autosubmit and Cylc exist. The issue of number of jobs is not the same as the issue of the queuing time for those jobs.

We thank the reviewer for their comments on the Abhinit et al. 2022 paper. We agree that our intention with the citation was unclear, therefore we removed it.

We have revised the introduction to include the aforementioned pilot-job systems, as well as other workflow managers that implement solutions similar to those of Autosubmit's wrappers and removed this reference. Our goal is to provide context for the reader and clarify that we do not claim novelty with wrapping.

The introduction, at lines 53-59, now reads as follows:

Aggregation was implemented in other fields, with different degrees of sophistication. In Earth Sciences, Mickelson et al. 2020 suggested using Cylc's feature to submit multiple jobs to reduce the queue time. Both Aiida (Huber et al. 2020) and Snakemake (Mölder et al. 2021)— from the material and life sciences fields, respectively — provide a way of to submit multiple workflow tasks as a single job. The former implements this via a ``metascheduler'' plugin (HyperQueue plugin), while the latter refers to aggregation as ``grouping.''

Moreover, Pilot-job systems were developed to increase workflow throughput, but later evolved into more sophisticated solutions (Turilli et al. 2018). These systems are characterized by implementing "resource placeholders, multi-level scheduling, and coordination patterns to enable task-level distribution and parallelism on multi-tenant

resources." One major example of a modern pilot-job system is Radical-PILOT (Merzky et al., 2021)

Wrappers share some of the features of a typical pilot-job system. Wrappers provide a simpler ``resource placeholder'', where all the task requests are added into a large submission. However, their objective is to either increase throughput by reducing submissions or to comply with maximum job restrictions. Therefore, besides fault tolerance, wrappers do not improve task scheduling and coordination within the allocation.

3. The decision to use pilot jobs for "wrappers" is not surprising, as pilot jobs have a long history, and a significant literature, none of which is referenced here.

We thank the reviewer for pointing out the absence of job-pilot systems in our contextualization. We have included an overview of pilot-job systems and cited a modern implementation of them in the introduction, which now reads as follows (lines 53–59):

Aggregation was implemented in other fields, with different degrees of sophistication. In Earth Sciences, Mickelson et al. 2020 suggested using Cylc's feature to submit multiple jobs to reduce the queue time. Both Aiida (Huber et al. 2020) and Snakemake (Mölder et al. 2021)— from the material and life sciences fields, respectively — provide a way of to submit multiple workflow tasks as a single job. The former implements this via a ``metascheduler'' plugin (HyperQueue plugin), while the latter refers to aggregation as ``grouping.''

Moreover, Pilot-job systems were developed to increase workflow throughput, but later evolved into more sophisticated solutions (Turilli et al. 2018). These systems are characterized by implementing "resource placeholders, multi-level scheduling, and coordination patterns to enable task-level distribution and parallelism on multi-tenant resources." One major example of a modern pilot-job system is Radical-PILOT (Merzky et al., 2021)

4. 7% is interesting, but they then say this corresponds to 8 days of their CMIP project, which means that they were running for 3-4 months. Saving eight days sounds less impressive in that context, and surely suggests on that timescale background workload would influence things by at least the same factor (that is our experience). The use of a short selected trace means this longer-term variability is not sampled, but the more important issue is the influence of JobSizeWeight and JobSizeFactor.

We thank the reviewer for pointing out the 7% gain and its implications for CMIP6 simulations. We agree that our explanation was not clear enough.

First, we would like to point out that aggregation is a simple and agnostic technique. It is independent of the underlying application, workflow manager, and platform. Contrast these gains with the cost-benefit of optimizing the code of these mature applications, which requires a lot of effort and platform-specific solutions.

Second, we would like to clarify that the 7% is the maximum difference between the time-to-solution of the unwrapped minus the wrapped workflow divided by the runtime of the workflow. We have observed across all the fair share values, that the wrapped workflow was shorter on average (in terms of time-to-solution) than its unwrapped counterpart, as indicated by the green triangle in Figure 6.

This takes us to the gains of using wrappers. These come from 1) jobs stay about the same or less in queue and 2) there are many times less jobs (if you wrap in groups of 10 a workflow with 50 sequential jobs, you would have 10 times less jobs submitted to the remote platform). Therefore, the longer the workflow and the wrappers, the larger should be the gains.

We clarified this in the manuscript by adding the rewriting the first paragraph of the discussion section, lines 266-273 . It now reads:

As seen in Figure 6, we achieved a reduction in queue time across all fair share values in the dynamic results. On average, this reduction was 1%, reaching up to a 7% decrease in queue time relative to the total workflow runtime. These results support the hypothesis that the reduction is caused by avoiding multiple submissions.

Since we observed consistent reductions when using aggregation, we anticipate greater gains in longer workflows because longer wrappers can be created, reducing the time spent waiting for resources.

Additionally, the 7% figure could be greater if we consider that the machine had only two days of congestion per week. Current flagship systems are usually congested, and it is not uncommon for jobs to queue for days.

Regarding the influence of JobSizeWeight and JobSizeFactor: we agree that we should be more precise about the behavior of all the scheduling factors. For this reason, we extended the "Scheduling Algorithm of Slurm" subsection to include the computation of all the factors and the final expression to compute the job's priority.

5. It is a pity that the discussion of horizontal wrappers was not followed through as that is likely to result in better throughput for ensembles where there is any risk of a failure during execution.

We greatly value the reviewer's comment about increasing throughput using horizontal wrappers. Therefore, we included it in the rewritten "Wrappers" subsection of the "Background" section (lines 116–126). It now reads:

In a shared HPC environment, queuing for resources is ever so frequent (Patel et al., 2020) and users have a limited impact on the priority of their jobs given the importance of fair share.

To reduce the time-to-solution of an Earth System Model (ESM) simulation workflow, the Autosubmit developers came up with a technique called task aggregation or wrapping. Their idea was to increase throughput by avoiding queuing subsequent tasks. For this reason they implemented vertical wrappers, which append workflow tasks into a longer submission.

In addition to vertical wrappers, horizontal wrappers were developed to comply with the platform's policy regarding the maximum number of jobs in the queue.

Finally, there is also the combination of the two types: vertical-horizontal and horizontal-vertical. A vertical-horizontal is made of multiple vertical wrappers running concurrently. Similarly, the horizontal-vertical is a single job made of multiple subsequent horizontal wrappers.

In all wrapper types, the dependencies among the tasks are respected and the underlying application task is not altered by their employment. Tasks are submitted together to the remote platform. Therefore, all steps normally performed, such as saving the restart conditions (or checkpointing), are still executed. Moreover, if a task fails within the aggregated job, Autosubmit will relaunch the failed task without the need of a new job submission.

In this work, we will focus on vertical wrappers, as they are the proposed solution for the long queue times.

6. The linear correlation exposed probably comes directly from the equation used by SLURM - surely they should include that equation and discuss the influence of all the key factors and relate to their results?

We thank the reviewer for their remark on the linear correlation. The reviewer's suspicion of this correlation is correct. This fact is explained by the weighted sum that Slurm uses to calculate the job priority.

To clarify this in the manuscript, we extended the section on Slurm's scheduling algorithms to explain how each factor is computed and how the job's priority is expressed (lines 85-94). It now reads:

When a job is submitted, Slurm immediately attempts to allocate resources to it. If no resources are available, the job will be placed in a queue. This queue is sorted in descending order according to the priority integer, which updates periodically.

Slurm's scheduling design consists of two coordinated algorithms that traverse the ordered list of queued jobs and attempt to allocate resources to them. The main algorithm tries to schedule as many jobs as possible from the queue. If it fails because of a lack of resources, it breaks the loop and sleeps. There are two options for the second algorithm: built-in and backfill. The first option works like the main algorithm. The second option is the backfill algorithm (Srinivasan et al., 2002).

The backfill algorithm is an optimization technique that allows jobs with lower priority to be scheduled before higher-priority jobs, as long as they do not interfere with the start time of the higher-priority jobs. Therefore, the smaller the job is in terms of wallclock and/or CPUs requested, the more likely it is to be backfilled. Without considering the effects of the backfill algorithm, jobs with higher priority are scheduled earlier.

Slurm uses the multifactor policy by default to compute the priority integer. In Slurm terminology, factors are always floats stored in double-precision between zero and one. The priority is computed as a weighted sum of the following factors: age, size, association, fair share, and quality of service (QoS). A 32-bit encoded integer weight is associated with each of these values by the configuration.

The age factor, which is the one related to the time in queue, starts at zero and linearly grows until reaching the maximum, which is configured by setting the flag PriorityMaxAge in the Slurm configuration file. The default is seven days. In Equation 1 we have the expression used to compute the factor where $t$ is the time elapsed from submission, discounted by the time voluntarily held by user, and $T$ is the total amount of seconds set by the administrator with flag PriorityMaxAge.

$$a_i(t) = \begin{cases} \frac{t}{T}, & t \leq T \\ 1, & t \geq T \end{cases}. \quad (1)$$

By default, the size factor is the proportion of CPUs used by the job with respect to the total available. Therefore, when the whole machine is requested, the maximum is reached. System administrators can invert this by setting the flag PriorityFavorSmall}to True, which would give job requesting a single CPU the maximum value.

The size factor is computed as in Equation 2, where $r_i$ is the number of CPUs requested by job $i$ and $S$ is the total number of CPUs on the machine.

$$s_i = \frac{r_i}{S} \ or \ s_i = \frac{S - r_i + 1}{S}. \quad (2)$$

QoS stands for quality of service. It is a mechanism that system administrators normally use to benefit smaller jobs (such as debug execution), since by default Slurm gives more priority to larger jobs. From the user side, this means higher priority upon submission, hence less waiting time, at the cost of a more constrained job submission. System administrators typically create different QoSs according to job length and size — for example, standard, debug, xlarge, and xlong.

Each QoS is configured with an associated priority, which is another integer value. Then the factor is computed as the proportion with respect to the largest priority of all QoSs. Therefore, if the job $i$ is submitted to QoS $u$ with priority $q_u$, its QoS factor is calculated according to Equation 3, where $Q$ is the set of the priorities of all QoSs configured by the system administrator.

$$q_i = \frac{q_u}{max_{q \in Q} q}.(3)$$

The fair share factor is the one that aims to balance the resources among users. It is computed in two different ways, which will be addressed in its separate Section 2.2.

Finally, the job's priority is computed according to Equation 4. $P_i(t)$ Is job $i$'s priority at time $t$. $a_i$ is the age factor, $w_a$ the age factor weight, $s_i$ is the size factor, $w_s$ the weight factor associated with size, $f_i$ is the fair share of the user that launched job $i$, $w_f$ is the weight, and finally $q_i$ and $w_q$ are the factor and weight associated to QoS.

$$P_i(t) = a_i(t) \cdot w_a + s_i \cdot w_s + f_i \cdot w_f + q_i w_q. \ (4)$$

We have also included the following comment about the linear relationship of the priority with respect to the fair share in the discussion (lines 287-288):

Finally, Figure 5 shows the expected linear relationship between priority and fair share. This is due to the weighted sum that Slurm uses to compute the priority in Equation 4. The slope of the fitted line, 84,374, nearly matches the fair share weight of the Slurm configuration that we used, which is 100,000.

---

## Author Response (AR1)

https://egusphere.copernicus.org/preprints/2025/egusphere-2025-1104#RC2

https://egusphere.copernicus.org/preprints/2025/egusphere-2025-1104#RC3

We would like to thank the reviewer for their prompt review. We address all of the comments below.

We did not understand the reviewer's recommendation for the manuscript to "be reorganized into five standard sections: Introduction, Data and Methods, Results and Analysis, Discussion, and Conclusion." We adopted a structure identical to the one recommended, with the only addition being the background section, which explains the fundamental relationship between scheduler factors and time in queue.

With regard to the "poorly structured" introduction not "clearly stating the research gap, objectives, and context within existing literature," we believe that all of the reviewer's concerns are addressed in the introduction. The research gap is stated on line 30, where we mention that "there has been a growing awareness of considering the entire execution of the workflow, taking into account not only the runtime of the most demanding part of it, but also the time spent queuing for resources and post-processing, with possible failures." Our objective is in lines 74-75 and also in the second to last paragraph of the introduction, where we say that "Our results help to advance the understanding, from the user side, on how to optimize the submission in order to reduce the total queue time of their workflows." Regarding the context within the literature, we state in lines 52–57 of the introduction that aggregation was identified elsewhere in the weather and climate community and that, as far as we know, there is no other work that tries to validate its usage.

As for the lack of a meaningful discussion section, "evaluating strengths and weaknesses, and situating the work in a broader scientific context." We believe we do address these points. For example, lines 268-269 draw attention to the relationship between a low fair share factor and aggregation improvement. We also reflect on the possible shortcomings of our methodology in lines 271-272, explaining that we rely on data from an old system that was not always as congested as current flagship systems. We also discuss the negative role of the backfill algorithm in lines 274-275. As for the broader scientific context, we remark — again — that this work is novel in the analysis within our context, as far as we know.

With regard to the reviewer's comment about the content being "overly simplistic, with limited methodological depth and superficial analysis," we would like to point out — again — that aggregation is used across various fields, including climate and weather, materials sciences (Aiida with HyperQueue [1]) and bioinformatics (Snakemake with grouping [2]). This work is therefore novel in its evaluation of this technique for solving the queue issue, which has never been tackled head-on in the literature. Therefore, we did it in the most direct and straight forward way.

As for our figures and quantitative results not providing "sufficient insight or generalizability for a scientific audience," we believe we were sufficiently general to cover modern HPC centers, given the available data, using the two distinct experiments. As stated in lines 301–303, "To have both modern job requests and realistic behavior on the usage of the machines, we performed two experiment types."

Finally, with regard to "the lack of rigorous validation or real-world deployment," we agree that real-world deployments would enrich our argument, executing them would require running multiple expensive concurrent simulations to test aggregation. Additionally, as Acosta et al. [3] have shown, the time in queue depends heavily on the specific platform. Therefore, we would also need to span this experimentation across sites. In conclusion, although we understand the request, we believe that real-world deployments are neither trivial nor cheap to run.

We value all reviews and comments, as we always strive to ensure that our science is as rigorous and accurate as possible. Therefore, we would now prefer to wait for the remaining reviews before deciding how to proceed. Thank you.

[1] https://aiida-hyperqueue.readthedocs.io/en/latest/

[2] https://snakemake.readthedocs.io/en/stable/executing/grouping.html

[3] https://gmd.copernicus.org/articles/17/3081/2024/

**Anonymous Referee #2**

https://egusphere.copernicus.org/preprints/2025/egusphere-2025-1104/#RC4

We would like to thank the reviewer for their constructive assessment of our submission. We will respond to each point in the order listed by the reviewer. We have highlighted the reviewer's comments in yellow and the changes made to the manuscript in response in blue.

(Technical report) The quantitative results are compelling and offer valuable insights for optimizing workflow submissions on congested HPC systems. However, there are still significant issues. I agree with another reviewer that the submission is much like a technical report, instead of a scientific paper. Further more, several technical problems should be addressed.

We appreciate the feedback and acknowledge that our submission appears to be a technical report. To address this, we revised the manuscript to provide more contextualization of our work within the current state of research and clarified the hypothesis.

Specifically, we expanded the introduction of this work to include a thorough state-of-the-art overview. The field of workflows is notoriously compartmentalized, but we have covered examples from multiple fields. Among the changes, we have included references to tools used in life and material sciences. Additionally, we have included pilot-job systems whose community will also benefit from this work because they also aggregate tasks into a large submissions. These changes were included in lines 53–59 and read as follows:

[revised manuscript text omitted]

In this work, we will focus on vertical wrappers, as they are the proposed solution for the long queue times.

Finally, we have further detailed our original hypothesis regarding the vertical wrappers in the introduction (lines 62–64). It now reads:

Although aggregation is utilized by Autosubmit users and also in other fields, with positive impact reported, there is a lack of understanding of the reasons and conditions under which aggregating reduces queue time. Therefore, in this work, we tested whether wrapping subsequent tasks together reduces queue time and if the fair share is the most important factor in reducing queue time.

And we clarified the abstract (lines 2-19). It now reads:

High Performance Computing (HPC) is commonly employed to run high-impact Earth System Model (ESM) simulations, such as those for climate change. However, running workflows of ESM simulations on cutting-edge platforms can take a long time due to the congestion of the system and the lack of coordination between current HPC schedulers and workflow manager systems (WfMS). The Earth Sciences community has estimated the time in queue to be between 10% to 20% of the runtime in climate prediction experiments, the most time-consuming exercise. To address this issue, the developers of Autosubmit, a WfMS tailored for climate and air quality sciences, have developed wrappers to submit multiple subsequent workflow tasks -- the indivisible unit of compute as defined by the workflow -- in a single remote job, without altering any of the tasks. However, although wrappers are widely used in production for community models such as EC-Earth3, MONARCH, and Destination Earth simulations, to our knowledge, the benefits and potential drawbacks have never been rigorously evaluated. Later, the developers of Autosubmit noticed that the impact of using wrappers was related to the past utilization of the user and its group, which the popular scheduler Slurm uses to compute the priority of the queueing jobs. In Slurm's methodology, this past utilization is quantified in the fair share factor. The objective of this paper is to quantify the impact of wrapping subsequent tasks on queue time and understand its relationship with the fair share and the job's CPU and runtime request. To do this, we used a Slurm simulator to reproduce the behavior of the scheduler and, to recreate a representative usage of an HPC platform, we generated synthetic static workloads from data of the LUMI supercomputer and a dynamic workload from a past flagship HPC platform. As an example, we introduced jobs modeled after the MONARCH air quality application in these workloads, and we tracked their queue time. We found that, by simply joining tasks, the total time-to-solution of the simulation reduces up

to 7% with respect to the runtime of the simulations, and we believe that this value is larger the longer the workflow, since longer wrappers could be created and hence less jobs are submitted to the scheduler. This saving translates to absolute terms of about eight days less wasted in queue time for half of the simulations from the IS-ENES3 consortium of CMIP6 simulations. We also identified in the static experiments a high inverse correlation of -0.87, between the queue time and the fair share factor.

(1) The manuscript states that the observed 7% reduction in runtime could be "larger in reality". Please expand on the specific real-world factors or complexities (e.g., more dynamic system loads, nuanced fair share policies, or varied backfill algorithm effectiveness) that might contribute to a greater benefit in practice. This would enhance the practical applicability and persuasiveness of the findings.

We thank the reviewer for this constructive comment, and we agree that we should explain precisely why we believe the 7% figure is likely higher in reality.

First, we would like to clarify that the 7% is the maximum difference between the time-to-solution of the unwrapped minus the wrapped workflow divided by the runtime of the workflow. We have observed across all the fair share values, that the wrapped workflow was shorter on average (in terms of time-to-solution) than its unwrapped counterpart, as indicated by the green triangle in Figure 6 that is always positive.

Thus the gains of using wrappers come from 1) the jobs stay about the same or less in queue and 2) there are many times less jobs (if wrappers of 10 tasks are employed in a workflow with 50 sequential jobs, there would be 10 times less jobs submitted to the remote platform).

Therefore, the longer the workflow and the wrappers, the larger should be the gains.

We have rewritten lines 266-273 that introduce the discussion with a clarification of the 7% and also why we believe that it would be beneficial, in general, for longer workflows.

As seen in Figure 6, we achieved a reduction in queue time across all fair share values in the dynamic results. On average, this reduction was 1%, reaching up to a 7% decrease in queue time relative to the total workflow runtime. These results support the hypothesis that the reduction is caused by avoiding multiple submissions.

Since we observed consistent (on average and on median) reductions when using aggregation, we anticipate greater gains in longer workflows running in congested environments because bigger wrappers can be created, reducing the number of submissions and therefore the time spent waiting for resources.

Additionally, the 7% figure could be greater if we consider that the machine had only two days of congestion per week. Current flagship systems are usually congested, and it is not uncommon for jobs to queue for days.

We observed three negative outliers, that is, the instances where the unwrapped workflow stood less in queue than the wrapped. We believe those were particular instances where the combination of running jobs and queuing jobs allowed the backfill algorithm to schedule the separate tasks earlier because they were smaller. Under this scenario, as we increase the length of the job, it is less likely that the scheduler would have free spots to execute, so the job has to wait to be scheduled by its priority. But, given the few instances where we observed arrangement, we believe this scenario is unlikely.

(2) While the Slurm simulator is a strength, a more explicit discussion of its known limitations and how these might influence the generalizability of the results is warranted. For instance, the paper mentions that the simulator "does not have support for dynamic submission times" for constrained jobs as a real Workflow Management System like Autosubmit would. While the authors address this by calculating submission times based on assumed predecessor completion, further detail on the potential implications of this approximation on the reported queue times would be beneficial.

We thank the reviewer for pointing out that the Slurm simulator is a strength of our methodology. We agree that we should be more explicit about its shortcomings.

Therefore, in line 294, we have included a new subsection called "Limitations" in the discussion section that reads:

In this subsection, we discuss the major weaknesses that we identified during our work.

First, the Slurm simulator does not support dynamic submissions (i.e., launching a job the moment its dependency finishes). Therefore, we had to define the submission time in advance by assuming the best-case scenario, in which no task is delayed. Thus, the age factor increases while the job waits in the queue for its dependency to finish, resulting in a higher priority than they would have in reality.

However, with the configuration we tested, we found that the priority added by the age factor was marginal. The maximum time that a job was in the queue in any of our simulations was 1,776 seconds, in the worst fair share case submitted at 15/6/2012 at 20 (submission instant B.3). This adds just 293 (1,176 seconds divided by the total number of seconds in seven days multiplied by $10^5$) to the job's priority. This is minimal compared with the priority added by a fair share of just 0.01, which, after being multiplied by its corresponding weight of $10^5$, adds 1,000 to the job's priority.

Another limitation is that the simulator does not support node sharing among jobs, as is the case in MareNostrum 4. Therefore less than a node requests would be scheduled to whole nodes, whereas, in MareNostrum 4, they would share resources. But, we have seen systems enforce node exclusivity across the board, as is the case with LUMI.

Finally, the Slurm simulator here employed is not determinist, although the authors of the BSC contribution to it greatly reduced it (Ana Jokanovic, Marco D'Amico, and Julita Corbalan, 2018).. This is another reason to run multiple experiments and take the average.

(3) The methodology for controlling fair share in static workloads using "dummy" jobs is clear. However, consider adding a brief discussion on whether this method fully captures the complex and dynamic evolution of fair share in a truly live, highly utilized HPC system.

We agree with the reviewer's suggestion to include a discussion of why "dummy" jobs capture the complex dynamics of HPC systems. We expanded the subsection that explains the experimental design of the static workloads (lines 197–201). The paragraph now reads:

In order to control the fair share, given that all the jobs of the workload are submitted at the same time, we preceded the simulation with a batch of ``dummy'' jobs so that all users have usage recorded. Otherwise, all users would have nil usage, therefore maximum fair share, and it would effectively remove the fair share from the scheduling. We set all the users ``dummy'' submission runtime to be proportional to the synthetically generated usage, except for the user employed with launching the workflow, for whom we chose its usage to control its fair share. This was done because we know that users have a recurrent pattern of utilization, as described by Patel et al. (2020). Therefore, the introduction of dummy jobs made the fair share of the users coherent with respect to the synthetically generated usage.

In the introduction to the static workloads, we also included a reference to a peer-reviewed paper utilizing static traces for performance modeling in HPC. Lines 170–172 now read:

Static workloads are those where all jobs in the description are submitted at the same time. We made this decision to simplify the modeling, since we disregard the complex modeling of the arrival time (Cirne and Berman, 2000).. We chose the number of jobs to be generated so that we stress the system but still within plausible bounds. This methodology was also employed by Jeannot et al. (2023) to recreate a HPC environment for performance analysis of their IO optimization technique.

(4) The paper outlines several categories of wrappers (vertical, horizontal, vertical-horizontal, and horizontal-vertical) but focuses solely on vertical wrappers. A brief justification for this specific focus, and perhaps a suggestion for future research avenues exploring the impact of the other wrapper types, would strengthen the introduction or discussion.

We agree that our explanation of why we focused on vertical wrappers was unclear. Therefore, we rewrote the entire subsection about wrappers in the background section (lines 116–126). It now reads:

In a shared HPC environment, queuing for resources is ever so frequent (Patel et al., 2020), and users have a limited impact on the priority of their jobs given the importance of fair share.

To reduce the time-to-solution of an Earth System Model (ESM) simulation workflow, the Autosubmit developers came up with a technique called task aggregation or wrapping. Their idea was to increase throughput by avoiding queuing subsequent tasks. For this

reason they implemented vertical wrappers, which append workflow tasks into a longer submission.

In addition to vertical wrappers, horizontal wrappers were developed to comply with the platform's policy regarding the maximum number of jobs in the queue.

Finally, there is also the combination of the two types: vertical-horizontal and horizontal-vertical. A vertical-horizontal is made of multiple vertical wrappers running concurrently. Similarly, the horizontal-vertical is a single job made of multiple subsequent horizontal wrappers.

In all wrapper types, the dependencies among the tasks are respected and the underlying application task is not altered by their employment. Tasks are just submitted together to the remote platform. Therefore, all steps normally performed, such as saving the restart conditions (or checkpointing), are still executed. Moreover, if a task fails within the aggregated job, Autosubmit will relaunch the failed task without the need of a new job submission.

In this work, we will focus on vertical wrappers, as they are the proposed solution for the long queue times.

**Anonymous Referee #3**

https://egusphere.copernicus.org/preprints/2025/egusphere-2025-1104/#RC5

We would like to thank the reviewer for their constructive comments. We will address each point raised by the reviewer in the original order and numbering. We have highlighted the reviewer's comments in yellow. Changes made to the manuscript in response are included in blue highlights.

**Major points:**

1. The material is not put well in the context of prior work, not only on measurement (e.g. Balazji et al 2017, whose results differ from those presented here - my quick calculation from their table 2 suggests the problem is of order 30%, so I don't know where the 10-20% they use as their motivation came from), and the large literature on pilot jobs on distributed and high throughput computing. The concept of wrapping things together is quite well established, but that leads me to my second concern:

We thank the reviewer for comment. We agree that the figure representing the overhead due to queue time was unclear. We have therefore changed the introduction (lines 30–35) to clarify it. It now reads:

However, lately the community has drawn attention to the efficiency of the simulations, taking into account not only the runtime of the simulation but also the time spent with the postprocessing, failures, and in the queue. With this in mind, Balaji et al. (2017) proposed a set of performance metrics for Earth system model simulations. Among these metrics, the authors proposed the simulated years per day (SYPD), which is the ratio of the time simulated in years with respect to the runtime of the simulation in days, as well as the actual simulated years per day (ASYPD), which is the simulated time in years divided by the the time-to-solution of the simulation, therefore also accounting for time in queue and failures.

In Acosta et al. (2024), the authors computed these metrics for 33 CMIP6 simulations executed on 14 machines. Their analysis showed that the difference between ASYPD and SYPD ranged from 0% to 78%. But, they noted that not all institutions reported ASYPD consistently. Some accounted for both interruptions and queue time, while others accounted only for queue time. For those institutions that only accounted for queue time, the spread was between 10% and 20%. The authors therefore concluded ``that queuing time represents an increment of around 10%–20% of the speed of the ESM.''

Regarding the contextualization of the work, we agree that pilot-job systems share a fundamental characteristic with wrappers, making them an essential topic for the introduction. Moreover, this community would also benefit of the results and discussions of our work.

In addition to the pilot-job systems, we have expanded the introduction to include other workflow managers that offer solutions similar to those of Autosubmit's wrappers. Lines 53-59 have been altered and now read:

Aggregation was implemented in other fields, with different degrees of sophistication. In Earth Sciences, Mickelson et al. (2020) suggested using Cylc's feature to submit multiple jobs to reduce the queue time. Both Aiida (Huber et al., 2020) and Snakemake (Mölder et al., 2021) — from the material and life sciences fields, respectively — provide a way of submitting multiple workflow tasks as a single job. The former implements this via a ``metascheduler'' plugin (HyperQueue plugin), while the latter refers to aggregation as ``grouping.''

Moreover, pilot-job systems were developed to increase workflow throughput with more sophisticated solutions (Turilli et al., 2018). These systems are characterized by implementing ``resource placeholders, multi-level scheduling, and coordination patterns to enable task-level distribution and parallelism on multi-tenant resources.'' One major example of a modern pilot-job system is Radical-PILOT (Merzky et al., 2021).

Wrappers share some of the features of a typical pilot-job system. Wrappers provide a simpler ``resource placeholder,'' where all the task requests are added into a large submission. However, their objective is to either increase throughput by reducing submissions or to comply with maximum job restrictions. Therefore, besides fault tolerance, wrappers do not improve task scheduling and coordination within the allocation.

2. One of the reasons that this is not done substantially in large-scale climate simulation is that the "chunks" of simulation (their nomenclature) provide natural checkpointing, and with bigger jobs than those they discuss model and hardware failures do lead to the need to checkpoint. Hardware failures are more prevalent now, and with bigger jobs that can be problematic. Perhaps this is why they only did Vertical Wrapping (they are only using one node per job if they only need 96 cores). Possibly horizontal wrapping would be more subject to this issue, but it might also be possible for the workflow manager to cope.

We thank the reviewer for the comment on checkpointing. We have expanded the introduction to clarify that the wrappers neither remove nor alter the frequency of the application's checkpoints.

Additionally, we explained the reasoning behind each type of wrapper (vertical and horizontal). We have revised the manuscript in the "Wrapper" subsection of the "Background" section (lines 116-126), that now reads:

In a shared HPC environment, queuing for resources is ever so frequent (Patel et al., 2020), and users have a limited impact on the priority of their jobs given the importance of fair share.

To reduce the time-to-solution of an Earth System Model (ESM) simulation workflow, the Autosubmit developers came up with a technique called task aggregation or wrapping. Their idea was to increase throughput by avoiding queuing subsequent tasks. For this reason they implemented vertical wrappers, which append workflow tasks into a longer submission.

In addition to vertical wrappers, horizontal wrappers were developed to comply with the platform's policy regarding the maximum number of jobs in the queue.

Finally, there is also the combination of the two types: vertical-horizontal and horizontal-vertical. A vertical-horizontal is made of multiple vertical wrappers running concurrently. Similarly, the horizontal-vertical is a single job made of multiple subsequent horizontal wrappers.

In all wrapper types, the dependencies among the tasks are respected and the underlying application task is not altered by their employment. Tasks are just submitted together to the remote platform. Therefore, all steps normally performed, such as saving the restart conditions (or checkpointing), are still executed. Moreover, if a task fails within the aggregated job, Autosubmit will relaunch the failed task without the need of a new job submission.

In this work, we will focus on vertical wrappers, as they are the proposed solution for the long queue times.

We rewritten the abstract to clarify that wrappers do not alter the underlying application (lines 2–19). The new text reads as follows:

High Performance Computing (HPC) is commonly employed to run high-impact Earth System Model (ESM) simulations, such as those for climate change. However, running workflows of ESM simulations on cutting-edge platforms can take a long time due to the congestion of the system and the lack of coordination between current HPC schedulers and workflow manager systems (WfMS). The Earth Sciences community has estimated the time in queue to be between 10% to 20% of the runtime in climate prediction experiments, the most time-consuming exercise. To address this issue, the developers of Autosubmit, a WfMS tailored for climate and air quality sciences, have developed wrappers to submit multiple subsequent workflow tasks -- the indivisible unit of compute as defined by the workflow -- in a single remote job, without altering any of the tasks. However, although wrappers are widely used in production for community models such as EC-Earth3, MONARCH, and Destination Earth simulations, to our knowledge, the benefits and potential drawbacks have never been rigorously evaluated. Later, the developers of Autosubmit noticed that the impact of using wrappers was related to the past utilization of the user and its group, which the popular scheduler Slurm uses to compute the priority of the queueing jobs. In Slurm's methodology, this past utilization is quantified in the fair share factor. The objective of this paper is to quantify the impact of wrapping subsequent

tasks on queue time and understand its relationship with the fair share and the job's CPU and runtime request. To do this, we used a Slurm simulator to reproduce the behavior of the scheduler and, to recreate a representative usage of an HPC platform, we generated synthetic static workloads from data of the LUMI supercomputer and a dynamic workload from a past flagship HPC platform. As an example, we introduced jobs modeled after the MONARCH air quality application in these workloads, and we tracked their queue time. We found that, by simply joining tasks, the total time-to-solution of the simulation reduces up to 7% with respect to the runtime of the simulations, and we believe that this value is larger the longer the workflow, since longer wrappers could be created and hence less jobs are submitted to the scheduler. This saving translates to absolute terms of about eight days less wasted in queue time for half of the simulations from the IS-ENES3 consortium of CMIP6 simulations. We also identified in the static experiments a high inverse correlation of -0.87, between the queue time and the fair share factor.

We have also changed the introduction (lines 48-51) to further emphasize that using wrappers does not affect the application.

Observing the impact of fair share on the priority, the developers of Autosubmit (Manubens-Gil et al., 2016), a Workflow Manager System (WfMS) designed for climate and air-quality applications, have implemented a task aggregation technique called ``wrappers.'' The idea is that Autosubmit aggregates, or wraps, multiple subsequent tasks into a single job submission to save the users from queuing multiple jobs under a low priority. No changes are done to the tasks, therefore the natural checkpointing is still maintained, and the dependencies are ensured among the tasks. Additionally, wrapping is also employed for concurrent tasks in order to comply with the maximum number of jobs in the queue.

3. The use of MONARCH and 96 core jobs (which are small for these systems) means that these results might not be typical of bigger jobs, not least because most large systems also prioritise bigger jobs, and so the fair-share factor influence may be less important. There is a large problem space that has not been examined. It's certainly the case that the CMIP extrapolation cannot be substantiated without exploring this in their simulations.

We thank the reviewer for the comment. We acknowledge that the jobs used in our experiments (96 cores) are small compared to typical CMIP simulations. And it is absolutely true that bigger jobs, in terms of cores, normally have more priority from the size factor. But our subject of study are the vertical wrappers, which were developed to avoid requeuing. Therefore we have clarified our hypothesis in lines 62-64 and now it reads:

[revised manuscript text omitted]

$$P_i(t) = a_i(t) \cdot w_a + s_i \cdot w_s + f_i \cdot w_f + q_i w_q.(4)$$

Finally, given the weights that size and fair share factors are normally assigned, we have that to match an extremely low fair share it would be necessary to make an extremely large request in CPU resources.

For example, in MareNostrum 5 — and it also applied for the previous version of the machine — the highest weight is the given to the QoS factor (10^6), followed by age and fair share (10^5), and finally size (10^4). Taking the largest simulation CMIP6 simulation analyzed by Acosta et al. 2024 in terms of CPUs, NERC-HadGEM3-GC3.1-HH, which requested 12,024 of the 118,080 available CPUs in the Archer x30 system, the size factor adds to the job a priority of 1,018 (12,024/118,080*10^4 = 1,018). This is nearly matched by an extremely low fair share of 0.01 (0.01*10^5 = 1,000). In the case of Destination Earth, their 30 year simulation runs on 305 nodes of the MareNostrum 5's general partition. This means the size factor adds to the job a priority of 470 (35,160/725,760*10^5 = 470) , as 35,160 out of the 725,760 available cores are requested. The aforementioned fair share of 0.01 (0.01*10^5 = 1,000) more than doubles the priority that the size factor gives to the job.

This argument is fundamentally dependent on the configuration of the system. What we observe is that the majority of the EuroHPC systems have weights that make the fair share more impactful on the job's priority than the size. In Lumi, both have the same weight (1,000), on MareNostrum fair share has ten times more impact (100,000 for fair share and

10,000 to size), and Leonardo is the only system that places a thousand times more weight to the size (10,000,000) versus the fair share (25,000).

**Minor points:**

1. There is a language (ASYPD, SYPD) for the difference between the overall throughput and the peak throughput that was introduced in Balaji et al (2017) - a paper that shares a co-author with this one, so it is surprising that language is not used, and that Balaji et al is not cited

We appreciate the suggestion to include the ASYPD and SYPD metrics since they are the standard for measuring the performance of climate and weather simulations. However, despite their strengths, they have an important temporal component, i.e., the total time simulated, which does not translate to other applications. For example, aggregation could be emplyed in genome sequence alignment applications typically executed with Snakemake [1] or material sciences simulations using SIESTA [2] with AiiDa [3]. Therefore, we prefer to continue using the universal metrics of time-to-solution and queue time, which applies across all fields.

But, following on the reviewer's comment, we have included Balaji et al (2017) reference to the manuscript, where we introduce the issue of queue time and climate simulations (lines 30-35). It now reads:

However, lately the community has drawn attention to the efficiency of the simulations, taking into account not only the runtime of the simulation but also the time spent with the postprocessing, failures, and in the queue. With this in mind, Balaji et al. (2017) proposed a set of performance metrics for Earth system model simulations. Among these metrics, the authors proposed the simulated years per day (SYPD), which is the ratio of the time simulated in years with respect to the runtime of the simulation in days, as well as the actual simulated years per day (ASYPD), which is the simulated time in years divided by the the time-to-solution of the simulation, therefore also accounting for time in queue and failures.

In Acosta et al. (2024), the authors computed these metrics for 33 CMIP6 simulations executed on 14 machines. Their analysis showed that the difference between ASYPD and SYPD ranged from 0% to 78%. But, they noted that not all institutions reported ASYPD consistently. Some accounted for both interruptions and queue time, while others accounted only for queue time. For those institutions that only accounted for queue time, the spread was between 10% and 20%. The authors therefore concluded ``that queuing time represents an increment of around 10%–20% of the speed of the ESM.''

[1] https://workflowhub.eu/workflows/547

[2] https://doi.org/10.1063/5.0005077

[3]
https://phantomsfoundation.com/AI4AM/2025/Abstracts/AI4AM2025_Garrido_Jaime_134.pdf

2. I had not seen Abhinit etal 2022, so I looked at it. I do not think it is saying the same thing as stated here. The problem in climate is unlikely to reach a need to wrap more than dozens of tasks (any more and the checkpoint issue dominates), whereas Abhinit et all were looking at wrapping thousands of tasks - in their case the issue is that most SLURM configurations do not have enough memory or resources to deal with the look ahead for queues with thousands of tasks. (Those that do are typically configured for High Throughput Computing, which is a different configuration to those encountered in most HPC sites where climate simulation is undertaken.) Abhinit et al's discussion is more relevant to Dask workflows than simulation workflows. That said, it is indeed the case that HPC sites often limit the number of jobs users can have in queues, which is why tools like Autosubmit and Cylc exist. The issue of number of jobs is not the same as the issue of the queuing time for those jobs.

We thank the reviewer for their comments on the Abhinit et al. 2022 paper. We agree that our intention with the citation was unclear, therefore we removed it.

We have revised the introduction to include the aforementioned pilot-job systems, as well as other workflow managers that implement solutions similar to those of Autosubmit's wrappers and removed this reference. Our goal is to provide context for the reader and clarify that we do not claim novelty with wrapping.

The introduction, at lines 53-59, now reads as follows:

Aggregation was implemented in other fields, with different degrees of sophistication. In Earth Sciences, Mickelson et al. (2020) suggested using Cylc's feature to submit multiple jobs to reduce the queue time. Both Aiida (Huber et al., 2020) and Snakemake (Mölder et al., 2021) — from the material and life sciences fields, respectively — provide a way of submitting multiple workflow tasks as a single job. The former implements this via a ``metascheduler'' plugin (HyperQueue plugin), while the latter refers to aggregation as ``grouping.''

Moreover, pilot-job systems were developed to increase workflow throughput with more sophisticated solutions (Turilli et al., 2018). These systems are characterized by implementing ``resource placeholders, multi-level scheduling, and coordination patterns to enable task-level distribution and parallelism on multi-tenant resources.'' One major example of a modern pilot-job system is Radical-PILOT (Merzky et al., 2021).

Wrappers share some of the features of a typical pilot-job system. Wrappers provide a simpler ``resource placeholder,'' where all the task requests are added into a large submission. However, their objective is to either increase throughput by reducing

submissions or to comply with maximum job restrictions. Therefore, besides fault tolerance, wrappers do not improve task scheduling and coordination within the allocation.

3. The decision to use pilot jobs for "wrappers" is not surprising, as pilot jobs have a long history, and a significant literature, none of which is referenced here.

We thank the reviewer for pointing out the absence of job-pilot systems in our contextualization. We have included an overview of pilot-job systems and cited a modern implementation of them in the introduction, which now reads as follows (lines 53–59):

Aggregation was implemented in other fields, with different degrees of sophistication. In Earth Sciences, Mickelson et al. (2020) suggested using Cylc's feature to submit multiple jobs to reduce the queue time. Both Aiida (Huber et al., 2020) and Snakemake (Mölder et al., 2021) — from the material and life sciences fields, respectively — provide a way of submitting multiple workflow tasks as a single job. The former implements this via a ``metascheduler'' plugin (HyperQueue plugin), while the latter refers to aggregation as ``grouping.''

Moreover, pilot-job systems were developed to increase workflow throughput with more sophisticated solutions (Turilli et al., 2018). These systems are characterized by implementing ``resource placeholders, multi-level scheduling, and coordination patterns to enable task-level distribution and parallelism on multi-tenant resources.'' One major example of a modern pilot-job system is Radical-PILOT (Merzky et al., 2021).

Wrappers share some of the features of a typical pilot-job system. Wrappers provide a simpler ``resource placeholder,'' where all the task requests are added into a large submission. However, their objective is to either increase throughput by reducing submissions or to comply with maximum job restrictions. Therefore, besides fault tolerance, wrappers do not improve task scheduling and coordination within the allocation.

4. 7% is interesting, but they then say this corresponds to 8 days of their CMIP project, which means that they were running for 3-4 months. Saving eight days sounds less impressive in that context, and surely suggests on that timescale background workload would influence things by at least the same factor (that is our experience). The use of a short selected trace means this longer-term variability is not sampled, but the more important issue is the influence of JobSizeWeight and JobSizeFactor.

We thank the reviewer for pointing out the 7% gain and its implications for CMIP6 simulations. We agree that our explanation was not clear enough.

First, we would like to point out that aggregation is a simple and agnostic technique. It is independent of the underlying application, workflow manager, and platform. Contrast

these gains with the cost-benefit of optimizing the code of these mature applications, which requires a lot of effort and platform-specific solutions.

Second, we would like to clarify that the 7% is the maximum difference between the time-to-solution of the unwrapped minus the wrapped workflow divided by the runtime of the workflow. We have observed across all the fair share values, that the wrapped workflow was shorter on average (in terms of time-to-solution) than its unwrapped counterpart, as indicated by the green triangle in Figure 6.

This takes us to the gains of using wrappers. These come from 1) jobs stay about the same or less in queue and 2) there are many times less jobs (if you wrap in groups of 10 a workflow with 50 sequential jobs, you would have 10 times less jobs submitted to the remote platform). Therefore, the longer the workflow and the wrappers, the larger should be the gains.

We clarified this in the manuscript by adding the rewriting the first paragraph of the discussion section, lines 266-273 . It now reads:

As seen in Figure 6, we achieved a reduction in queue time across all fair share values in the dynamic results. On average, this reduction was 1%, reaching up to a 7% decrease in queue time relative to the total workflow runtime. These results support the hypothesis that the reduction is caused by avoiding multiple submissions.

Since we observed consistent (on average and on median) reductions when using aggregation, we anticipate greater gains in longer workflows running in congested environments because bigger wrappers can be created, reducing the number of submissions and therefore the time spent waiting for resources.

Additionally, the 7% figure could be greater if we consider that the machine had only two days of congestion per week. Current flagship systems are usually congested, and it is not uncommon for jobs to queue for days.

We observed three negative outliers, that is, the instances where the unwrapped workflow stood less in queue than the wrapped. We believe those were particular instances where the combination of running jobs and queuing jobs allowed the backfill algorithm to schedule the separate tasks earlier because they were smaller. Under this scenario, as we increase the length of the job, it is less likely that the scheduler would have free spots to execute, so the job has to wait to be scheduled by its priority. But, given the few instances where we observed arrangement, we believe this scenario is unlikely.

Regarding the influence of JobSizeWeight and JobSizeFactor: we agree that we should be more precise about the behavior of all the scheduling factors. For this reason, we extended the "Scheduling Algorithm of Slurm" subsection to include the computation of all the factors and the final expression to compute the job's priority.

5. It is a pity that the discussion of horizontal wrappers was not followed through as that is likely to result in better throughput for ensembles where there is any risk of a failure during execution.

We greatly value the reviewer's comment about increasing throughput using horizontal wrappers. Therefore, we included it in the rewritten "Wrappers" subsection of the "Background" section (lines 116–126). It now reads:

In a shared HPC environment, queuing for resources is ever so frequent (Patel et al., 2020), and users have a limited impact on the priority of their jobs given the importance of fair share.

To reduce the time-to-solution of an Earth System Model (ESM) simulation workflow, the Autosubmit developers came up with a technique called task aggregation or wrapping. Their idea was to increase throughput by avoiding queuing subsequent tasks. For this reason they implemented vertical wrappers, which append workflow tasks into a longer submission.

In addition to vertical wrappers, horizontal wrappers were developed to comply with the platform's policy regarding the maximum number of jobs in the queue.

Finally, there is also the combination of the two types: vertical-horizontal and horizontal-vertical. A vertical-horizontal is made of multiple vertical wrappers running concurrently. Similarly, the horizontal-vertical is a single job made of multiple subsequent horizontal wrappers.

In all wrapper types, the dependencies among the tasks are respected and the underlying application task is not altered by their employment. Tasks are just submitted together to the remote platform. Therefore, all steps normally performed, such as saving the restart conditions (or checkpointing), are still executed. Moreover, if a task fails within the aggregated job, Autosubmit will relaunch the failed task without the need of a new job submission.

In this work, we will focus on vertical wrappers, as they are the proposed solution for the long queue times.

6. The linear correlation exposed probably comes directly from the equation used by SLURM - surely they should include that equation and discuss the influence of all the key factors and relate to their results?

We thank the reviewer for their remark on the linear correlation. The reviewer's suspicion of this correlation is correct. This fact is explained by the weighted sum that Slurm uses to calculate the job priority.

To clarify this in the manuscript, we extended the section on Slurm's scheduling algorithms to explain how each factor is computed and how the job's priority is expressed (lines 85-94). It now reads:

[revised manuscript text omitted]

We have also included the following comment about the linear relationship of the priority with respect to the fair share in the discussion (lines 287-288):

Finally, Figure 5 shows the expected linear relationship between priority and fair share. This is due to the weighted sum that Slurm uses to compute the priority in Equation 4. The slope of the fitted line, 84,374, nearly matches the fair share weight of the Slurm configuration that we used, which is 100,000.